



# Arctic glacier snowline altitudes rise 150 meters over the last four decades

Laura J. Larocca[1,2,3], James M. Lea[4], Michael P. Erb[1], Nicholas P. McKay[1], Megan Phillips[1], Kara A. Lamantia[5,6], Darrell S. Kaufman[1]

5   [1]School of Earth and Sustainability, Northern Arizona University, Flagstaff, AZ 86011, USA

[2]The Cooperative Programs for the Advancement of Earth System Science, University Cooperation for Atmospheric Research, Boulder, CO 80307, USA

[3]School of Ocean Futures, Arizona State University, Tempe, AZ 85281, USA

[4]Department of Geography and Planning, University of Liverpool, Liverpool, UK

10  [5]Byrd Polar and Climate Research Center, Ohio State University, Columbus, OH, USA

[6]School of Earth Sciences, Ohio State University, Columbus, OH, USA

*Correspondence to*: Laura J. Larocca (LJLarocc@asu.edu)

**Abstract.** The number of Arctic glaciers with direct, long–term measurements of mass balance is limited. Here we used satellite–based observations of the glacier snowline altitude (SLA), the location of the transition between snow cover and ice late in the summer, to approximate the position of the equilibrium line altitude (ELA)–a parameter important for mass balance assessment and for understanding the response of glaciers to climate change. We mapped the snowline (SL) on a subset of 269 land–terminating glaciers above 60 °N latitude in the latest available summer, clear–sky Landsat satellite image between 1984 and 2022. The mean SLA was extracted using the Advanced Spaceborne Thermal Emission and Reflection Radiometer (ASTER) Global Digital Elevation Model (GDEM). We compare remotely–observed SLA observations with available long–term field–based measurements of ELA and with ERA5–Land reanalysis climate data. Over the last four decades, Arctic glacier SLAs have risen an average of ~152 m (3.9±0.4 m yr$^{-1}$; R$^2$=0.74, p<0.001), with a corresponding summer (June, July, August) temperature shift of +1.2 °C at the glacier locations. This equates to a 127±5 m shift per 1 °C of summer warming. However, along with warming, we observe an overall decrease in snowfall, an increase in rainfall, and a decrease in the total number of days in which the mean daily temperature is less than or equal to 0 °C. Glacier SLA is most strongly correlated with the number of freezing days, emphasizing the dual effect of multi–decadal trends in mean annual temperature on both ablation (increasing melt) and accumulation processes (reducing the number of days in which snow can fall). Although we find evidence for a negative morpho–topographic feedback that occurs as glaciers retreat to higher elevations, we show that more than 50% of the glaciers studied here could be entirely below the SLA by 2100, assuming the pace of global warming and the mean rate of SLA rise is maintained.



## 1 Introduction

Glaciers distinct from Earth's ice sheets are rapidly receding with wide ranging effects on water resources, regional hydrology, natural hazards, and sea–level (e.g., Hock et al., 2019). Over the past two decades, their mass loss has constituted 21% of the observed sea–level rise (Hugonnet et al., 2021). Roughly a quarter of Earth's glaciers, which account for ~60% of the global total glacierized area, lie in the Arctic—a region that has warmed almost four times faster than the globe since 1979 (RGI Consortium, 2017; Rantanen et al., 2022). Owing to their broadly dispersed, remote, and logistically challenging settings, direct field observations on glaciers are sparse. Although glaciological mass–balance observations from ~480 glaciers have been collected and are available at the World Glacier Monitoring Service (WGMS), only ~42 glaciers worldwide have continuous records spanning more than 30 years (WGMS, 2021). This not only hinders our understanding of glacier response to climate variability and change, including regional response to large–scale atmospheric circulation patterns, but also limits the temporal context for very recent changes (e.g., Bjørk et al., 2012, 2018; Larocca et al., 2023). Further, long–term, multi–decade, observations of glacier change have also proven invaluable for comparison to model projections of twenty–first century glacier mass loss and contribution to sea–level rise (Khan et al., 2020).

Where field–based measurements are lacking, satellite–based observations of the snowline altitude (SLA; the location of the transition between snow cover and ice at the end of the ablation season) can be used to approximate the glacier equilibrium–line altitude (ELA; e.g., Braithwaite, 1984; Pelto, 2011; Rabatel et al., 2005, 2013; Mernild et al., 2013). The ELA marks the average elevation of the transitional line or zone separating the accumulation area of a glacier from the ablation area, at which the annual net mass balance is zero (Meier, 1962; Braithwaite and Müller, 1980). Therefore, there is a close relationship between ELA and glacier mass balance, as they are directly related to local climate variables that control accumulation and ablation processes, namely solid precipitation and summer season air temperature. Thus, fluctuations in the ELA provide an important indicator of glacier response and sensitivity to climate shifts. By utilizing long–operational satellite missions, such as the Landsat program, glacier SLA and thus ELA can be reconstructed over extended periods of time, and over large spatial scales. This is particularly useful in regions where multi–decade observations are especially lacking, such as in many regions of the Arctic. For example, in Greenland, there is only one glacier out of roughly twenty thousand with prolonged field–based measurements of mass balance, which begin from 1995 (Mernild et al., 2011).

Here we present late summer (July–September) SLA timeseries for 269 land–terminating glaciers located above 60 °N latitude over a nearly 40–year period. Specifically, we digitized the positions of 3489 snowlines between 1984 and 2022 using optical imagery from the suite of Landsat satellites and extracted their mean elevations using the Advanced Spaceborne Thermal Emission and Reflection Radiometer Global Digital Elevation Model (ASTER GDEM; **Fig. 1**). We also compared the SLA observations with long–term ELA measurements where available, with ERA5–Land reanalysis climate data from the European Centre for Medium–Range Weather Forecasts (ECMWF), and with glacier morpho–topographic parameters. We aim (1) to




quantify change in glacier SLA broadly (i.e., Arctic–wide); (2) to identify spatial variations in the rate of glacier SLA change across the Arctic's diverse climate zones; and (3) to characterize the relationships between glacier SLA change, climate and
morpho–topographic variables. Thus, this observation–based study can be compared with models and theory, including glacier sensitivity to temperature versus precipitation, and with future projections of glacier loss.

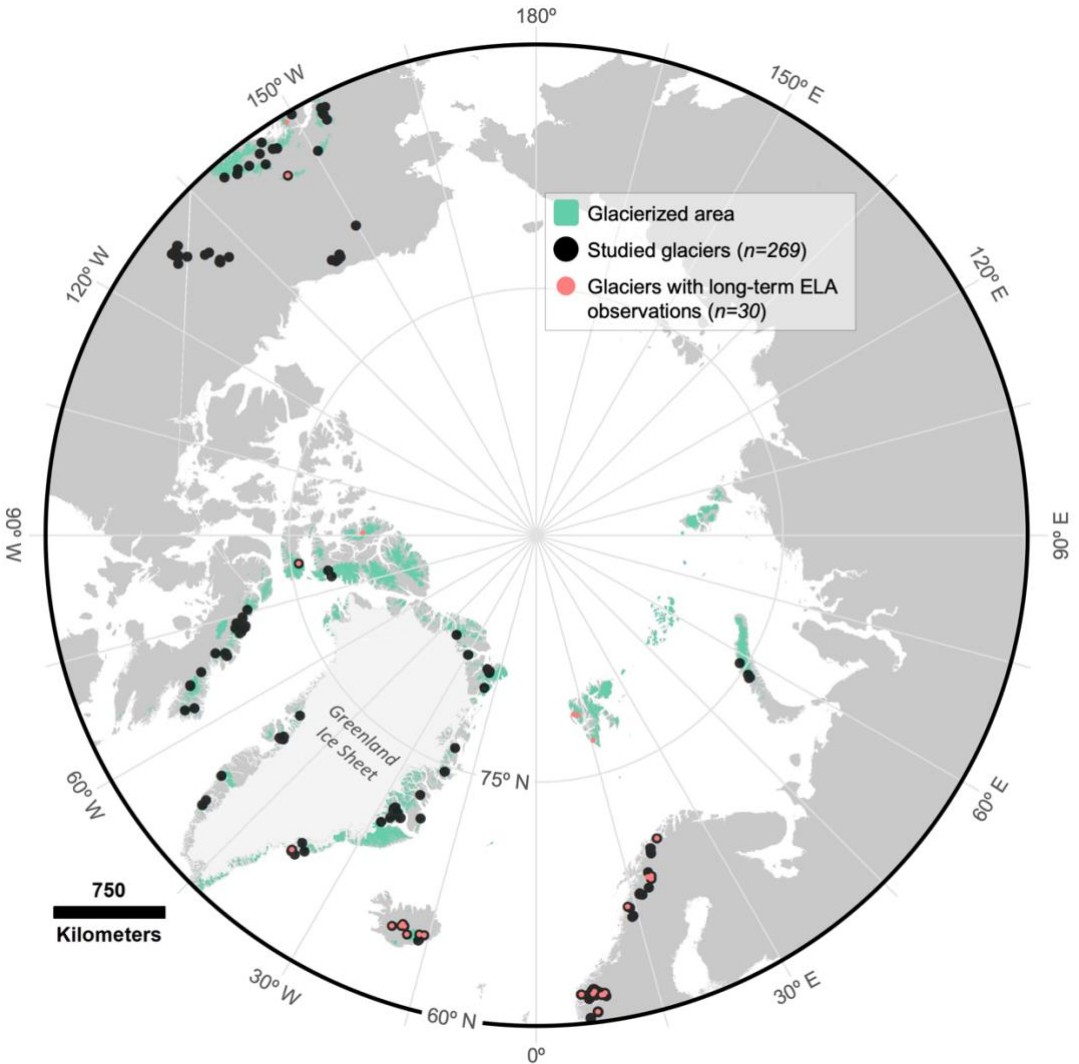

**Figure 1.** Map of the Arctic and glacier locations. Glacierized area distinct from the Greenland Ice Sheet is shown in green;
the locations of the 269 land–terminating glaciers included in this study for which snowlines were digitized and analysed are denoted by black circles; and the locations of the 30 glaciers with long–term annual observations of equilibrium–line altitude (ELA) are denoted by smaller pink circles. Glacierized area from the Randolph Glacier Inventory (RGI) Version 6 (RGI Consortium, 2017).



## 2 Data and methods

### 2.1 Selection of study sites and glacier snowline delineation

More than 50,000 glaciers sit above 60 °N latitude (**Fig. 1**; RGI Consortium, 2017). We documented SLA for glaciers with available long–term reconstructions, as well as those with long–term annual mass balance and ELA observations (i.e., Larocca and Axford, 2022; Ohmura and Boettcher, 2022; *n=227*). To expand and broaden our dataset, we selected an additional set of glaciers in which to document SLA from the ten first–order regions defined by the Randolph Glacier Inventory (RGI) that intersect the Arctic (*n=148*). All measured glaciers fit within the following criteria: the glacier (1) terminates on land; (2) is in the Arctic (which we define as land area above 60 °N latitude); and (3) is not surging (or has no record of surging behavior). The aforementioned field–based glaciological data represent a global dataset of quality checked, extended ELA records compiled from multiple sources including the World Glacier Monitoring Service (WGMS) and other national data sources (as well as estimated missing yearly ELA values computed by correlation with annual net mass balance; Ohmura and Boettcher, 2022). The dataset includes ELA observations for 70 glaciers, 30 of which are in Arctic regions (**Fig. 1**). The 30 glaciers included in our analysis have at least two decades of ELA observations, ranging from 21 to 74 years of observation (**Table S1**). As these data were used to assess how well the remotely documented late summer glacier SLA represent the field–based observations of ELA, we included only the direct measurements and removed any estimated values of ELA from our analysis.

To digitize the position of the glacier SL, we used the *Google Earth Engine Digitisation Tool* (GEEDiT), which allows for rapid access to, and visualization of the full Landsat image collection, as well as rapid mapping of georeferenced vectors that can be exported with image metadata (Lea et al., 2018). Although there has been progress in recent years in the development of novel automated methods that distinguish spectrally similar snow from ice on glaciers (e.g., Rastner et al., 2019; Racoviteanu et al., 2019; Li et al., 2022), these methods can be impeded by clouds, topography, and their shadowing, and most still require download, storage, and processing of a large amount of data. Thus, to ensure that SL observations were not erroneously identified due to these complicating factors, we digitized all glacier SLs manually. SLs were delineated as the location of the transition from fresh snow cover (or firn) to bare ice across the width of the glacier on true color Landsat 4−9 images as close as possible to the end of the hydrological year, and before any late summer snowfall events (July–September; **Fig. 2a** and **Table 1**). SLs were digitized on cloud–free scenes only when a clear visual boundary between snow cover and ice could be confidently identified. As such, digitization of the position of the SL was not possible for all years and glaciers for two main reasons: (1) clear–sky imagery at the end of the ablation season is sometimes not available; and (2) in some images, terrain shadowing obscures the location of the SL. In addition, in cases where the entire glacier is ice–covered (i.e., the snowline is above the glacier surface), we did not digitize the SL. Thus, the total number of SLA observations collected vary by year (**Fig. S1**).



To extract the mean altitude of the glacier SLs, we overlayed the georeferenced vector data output from GEEDiT on ASTER GDEM V3 mosaiced tiles and used the ArcGIS tool, *Add Surface Information* (which attributes features with spatial information derived from a surface), to compute the average surface elevation along the length of each SL (**Fig. 2b**). The
110 ASTER GDEM V3 has a spatial resolution of 30 m and a vertical accuracy of 16.7 m at a 95% confidence level (Gesch et al., 2016). While the vertical accuracy is indicative of the potential error in the absolute position of the SLAs, it is not directly indicative of the error in the relative change in SLA over time.

Next, we selected a subset of the initial set of 375 glaciers with adequate temporal coverage of SLA over the 39–year
observational period based on the following requirements: we selected glaciers with five or more total SL observations over the observational period (i.e., between 1984–2022); at least one observation in each third of the observational period (i.e., in the first, middle, and last 13 years); and a maximum gap of 15 years between SL observations. The resulting subset (*n=269*; **Fig. 1**) represents ~0.5% of the entire Arctic glacier population, and includes glaciers from eight of the ten first–order regions, defined by the RGI, that intersect the Arctic (RGI Consortium, 2017). We used this subset of glaciers in the subsequent analyses
and assessed the representativeness of this sample with respect to all land–terminating glaciers in the Arctic in terms of climate setting and morpho–topographic characteristics (**Table 2 and Fig. S2**).

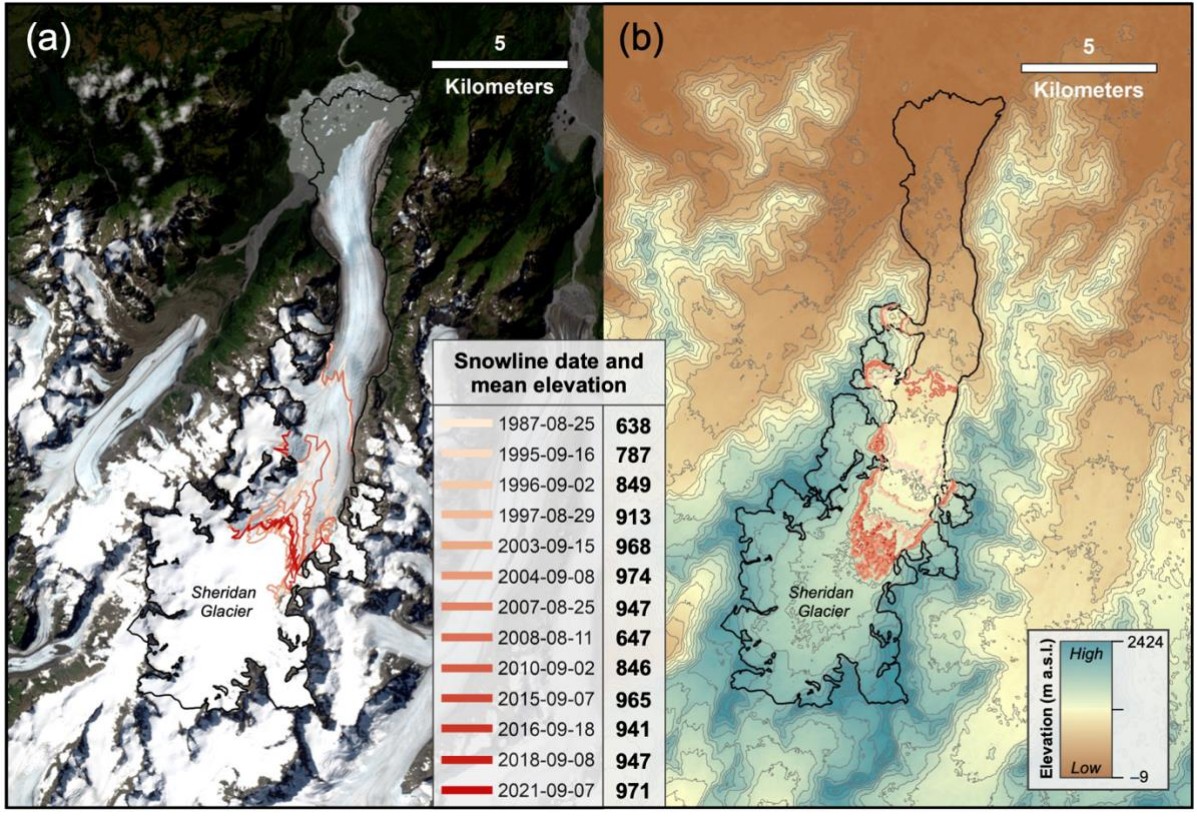



**Figure 2.** Sheridan Glacier (ID: RGI60–01.01854) in the Chugach Mountains of Alaska. **(a)** Landsat–8 satellite image from late summer (Sept. 7) 2021 (USGS Earth Explorer). The position of the snowline (SL) is shown during 13 years from beige (1987) to red (2021). **(b)** ASTER Global Digital Elevation Model (GDEM) Version 3 (NASA Earth Data). The mean elevation of each SL as defined in (a) is represented by its contour (and colored by date) from the ASTER GDEM. Light gray lines are 100 m contours. Glacier extent (black polygon) in both panels is from the Randolph Glacier Inventory (RGI) Version 6.

**Table 1.** Description of satellites used to delineate glacier snowlines.

| Satellite | Imagery type | Lifespan | True color bands (R–G–B) | Spatial resolution (m) |
|---|---|---|---|---|
| Landsat 4 | Optical | 1982–1993 | 3–2–1 | 30 |
| Landsat 5 | Optical | 1984–2013 | 3–2–1 | 30 |
| Landsat 7* | Optical | 1999– | 3–2–1 | 30 |
| Landsat 8 | Optical | 2013– | 4–3–2 | 30 |
| Landsat 9 | Optical | 2021– | 4–3–2 | 30 |

*Scan Line Corrector failure after May 2003

## 2.2 Normalization and compositing of snowline altitude observations

To mitigate issues related to combining timeseries from multiple glaciers that include years with missing SLA observations in our dataset, we computed normalized SLA timeseries expressed in units of standard deviation (SD) relative to a reference period or climatological normal (e.g., Lorrey et al., 2022). For each glacier location, we first computed the mean and SD of the SLA between 1984–2022 (i.e., using the full observational period as the reference period). Next, we computed a standard z–score (z) for each yearly SLA observation:

$$z = (x–\mu)/\sigma, \tag{1}$$

where x is the SLA, μ is the mean SLA during the reference period, and σ is the SD during the reference period.

To account for the uneven spatial distribution of glaciers included in our analysis and to mitigate the influence of clustered sites (**Fig. 1**), SLA timeseries were averaged within equal–area grids of 10,000 km². This results in 70 total grid cells with an average of 4±9 (1 SD) glaciers that fall within each (**Fig. 3**). For each year in the observational period (1984–2022), we computed a mean z–score for each grid cell, and a composite value representing the mean z–score across all grid cells. This results in a normalized composite time series that represents the average Arctic–wide glacier SLA change, and the ± 1 SD characterizes the variability in SLA across grid cells in each year.





**Figure 3.** Maps of 10,000 km$^2$ grid cells with glacier snowline observations from this study. **(a)** Glacier count per grid cell. Color corresponds to the number of glaciers that fall within each grid cell. **(b)** Grid cell identifiers for Alaska, **(c)** Scandinavia, **(d)** Arctic Canada, Greenland, Iceland, and **(e)** the Russian Arctic, numbered 1–70. Maps are in Lambert azimuthal equal–area projection.

## 2.3 Climate observations and glacier morpho–topographic variables

ERA5–Land monthly averaged and hourly data over the observational period were obtained from the Copernicus Climate Change Service (C3S) Climate Data Store (CDS; Muñoz, 2019). The native spatial resolution of the ERA5–Land reanalysis dataset is 9 km, however the data in the CDS is regridded to a regular latitude–longitude grid of 0.1 x 0.1 degrees (Muñoz,





2019). To characterize glacier–climate relationships, we focused on the following climate variables: temperature, snowfall,

total precipitation, and rainfall (total precipitation minus snowfall) and extract information for each of the 269 glaciers, at their specific latitude–longitude locations using a nearest neighbour approach. For each glacier location, annual and seasonal mean values were computed for temperature in degrees Celsius, and annual and seasonal total values were computed for precipitation variables in mm water equivalent (mm w.e.). One potential caveat of using precipitation reanalysis products is that they are generally more uncertain in areas of mountainous or highly complex terrain, especially regarding orographically induced

precipitation (e.g., Davaze et al., 2020; Hamm et al., 2020; Gomis–Cebolla et al., 2023). In addition, we computed freeze days (i.e., the total number of days in which the mean daily temperature is less than or equal to 0 °C) and the positive degree day (PDD) sum (i.e., the sum of the mean daily temperature above 0 °C; Cogley et al., 2011), annually and seasonally. For equal comparison to our SLA dataset, we computed yearly normalized climate observations (relative to the climatological reference 1984–2022), along with gridded composite time series following the methods outlined in Section 2.2.

Glacier morphology can be indicative of local microclimate factors that could influence SLA, such as sheltering from solar radiation within high mountain cirques, wind–blown snow accumulation, and avalanching. Thus, we also extracted several glacier–specific morpho–topographic variables (minimum, median, and maximum elevation; area; length; aspect; and slope) directly from the RGI V6.0, computed perimeter using glacier outlines as defined by the RGI, and computed a measure of

175 compactness as follows (e.g., DeBeer and Sharp, 2009; Way et al., 2014):

$$\text{Compactness} = 4\pi A/P^2, \tag{2}$$

where A is the area and P is the perimeter of the glacier. A score of zero indicates a complete lack of compactness and a score of one indicates maximal compactness (i.e., a circle).

**3 Results**

**3.1 Evaluating glacier snowline altitude as a proxy for equilibrium–line altitude**

We find a robust relationship between the remotely–observed SLAs and the field–measured ELAs for the 30 glaciers with long–term, quality observations (**Fig. 4a**; $R^2$=0.92, p<0.001). This suggests that the late summer SLA is a good proxy for ELA for glaciers in our study region. However, when accounting for the elevation of each glacier, the satellite observed SLA is

185 generally below (mean (M)=–106 m) the corresponding ELA for glaciers in most regions (**Fig. 4b**). This is likely because the satellite image acquisition date of the SLA observations may be prior to the very end of the ablation season. Transient SL (TLS) data from two Arctic glaciers suggest that the rate of SL rise over the ablation season is on par with the difference between our remotely–observed SLAs and the field–measured ELAs (for example, the measured ablation season mean TSL elevation rate of ~3.8–9.4 m per day translates to an additional ~56–132 m rise if the SLA observation was two weeks prior to

190 the end of the summer season; Mernild et al., 2013). This also suggests that, in general, superimposed ice (the refreezing of




meltwater at the glacier ice surface) is not problematic, as we would expect the remotely–observed SLAs to be systematically higher than the field–measured ELAs if superimposed ice zones were consistently missed in the optical imagery (see Discussion section 4.1). Although our remotely observed SLAs may be underestimating the end of season ELA, the mean rate of SLA and ELA change across all 30 glaciers are comparable over the observational period: $4.0\pm3.5$ m yr$^{-1}$ and $3.3\pm2.8$ m yr$^{-1}$, respectively (ELA observations from Ohmura and Boettcher, 2022).

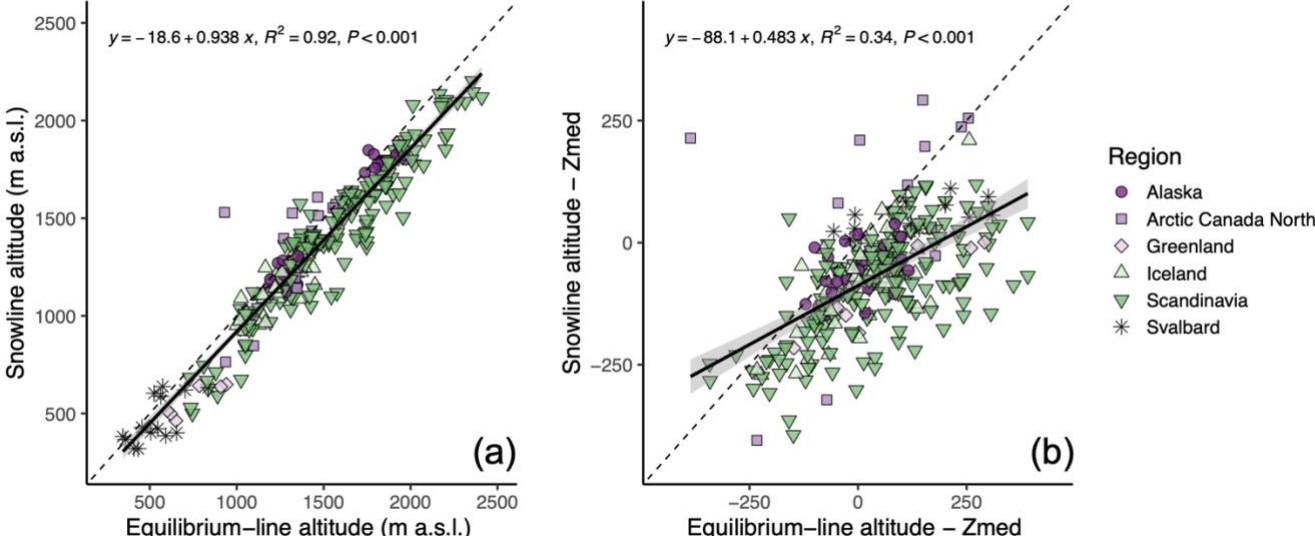

**Figure 4.** Glacier snowline altitude as a proxy for equilibrium–line altitude **(a)** Relationship between the yearly observed snowline altitude and the measured equilibrium–line altitude for 30 glaciers with long–term measurements within six regions (see **Table S1** for individual glacier information; glacier ELA timeseries was compiled by Ohmura and Boettcher, 2022). **(b)** Snowline altitude observations and the equilibrium–line altitude measurements minus the median elevation ($Z_{med}$) of each glacier. In both panels, the dashed reference lines have a slope of 1, and linear regression models along with 95% confidence intervals (gray bands) are overlayed.

## 3.2 Representativeness of the selected glaciers

In terms of morpho–topographic characteristics, in general our subset of glaciers are larger and longer, and are less steep, as compared to the average across all land–terminating Arctic glaciers in the RGI ($n=50,490$; **Fig. S2**). Observing snowline on very small glaciers can be difficult given Landsat's relatively coarse spatial resolution, and therefore the size (area and length) of the glaciers in our sample are, on average, larger than compared to the Arctic average (~74% of the Arctic glacier population are < 1 km$^2$ in area; **Fig. S2**). Climatologically, the overall trends in temperature and precipitation are comparable in direction but vary in magnitude across our selected glacier sites, all glaciated areas in the Arctic, and the Arctic region as a whole (including all land area above 60 °N; **Table 2**). Specifically, our sampled glaciers have experienced somewhat less



warming than the RGI and Arctic means, annually and across all seasons. This may be attributed to the underrepresentation of studied glaciers at very high latitudes and in the Russian Arctic, where warming has been more pronounced. In comparison to the RGI and Arctic averages, our studied glaciers have also experienced a larger decrease in snowfall

annually, and a larger increase in rainfall annually and in summer, fall and spring. Further, total precipitation in summer has also significantly increased (likely reflecting the increase in rain) at the studied glaciers, whereas there are no significant trends in total precipitation across the RGI glaciated areas, or across the Arctic as a whole. The change in number of freezing days is comparable between the studied glaciers, and the RGI and Arctic means. However, the Arctic mean shows a higher magnitude increase in the positive degree day sum annually, and in summer, fall and spring, as compared to the studied

glacier locations and RGI mean.

### 3.3 Arctic–wide change in glacier snowline altitude

The mean SLA for the 1984–2022 reference period is located at ~1267±84 m a.s.l., considering glaciers within all grid cells. Overall, we observe a rise in glacier SLA over the 39 years of observation (**Fig. 5**). A linear regression model, using the yearly mean composite values, suggests an average rise in SLA of 3.9±0.4 m yr$^{-1}$ (R$^2$=0.74, p<0.001), which corresponds to an average

increase of 152.1±2.5 m (or 1.8 SD) over the 1984–2022 period. Notably, this change is roughly three times the interannual variability of the mean yearly values, SD=52 m. We also observe substantial spatial variation across grid cells in the mean rate of snowline altitude change over the 39–year period, ranging between –2.6 and +15.2 m yr$^{-1}$ (M=4.3 m yr$^{-1}$; SD=3.3 m yr$^{-1}$; **Fig. 5a**). We explore this spatial heterogeneity in the next sections.





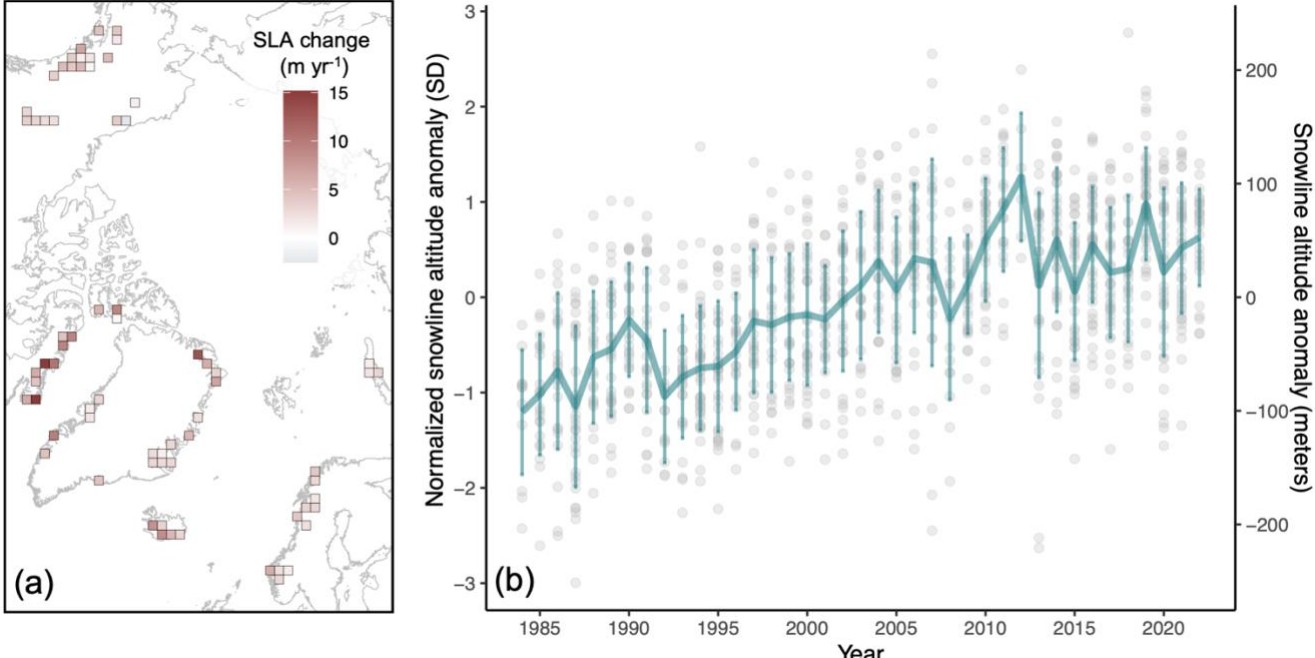


**Figure 5.** Arctic–wide change in snowline altitude over 39 years. **(a)** Mean rate of snowline altitude change (meters per year) for glaciers within each grid cell between 1984–2022. Map is in Lambert azimuthal equal–area projection. **(b)** Composite time series of snowline altitude from 1984–2022. Teal line is the mean of all grid cells. Uncertainty is represented as the spread, equivalent to 1 SD, of all grid cells (gray points) that contribute to the mean value for each year. Right–side y–axis is based

on a conversion of 1 SD=84 m (1 SD equivalence in meters was computed by taking the mean of the baseline SD for glaciers within each grid cell, and across grid cells). Data are relative to the 1984–2022 mean.

**3.4 Spatial variability in glacier snowline altitude change**

We summarized the spatial variation in glacier SLA change across the Arctic by highlighting two quartile groups: grid cells

that fall into the *highest* (*n=18*) and the *lowest* 25% (*n=18*) of the distribution of the rate of SLA change (**Fig. 6a; Fig. S3**). In general, glaciers located in Arctic Canada (proximal to Baffin Bay), northern Greenland and western Iceland show the highest average rate of SLA change (**Fig. 5a and 6a**). The mean rate of glacier SLA change across grid cells within the *highest* group is 9.0 m yr$^{-1}$ (SD=3.4 m yr$^{-1}$). Conversely, in general glaciers located in some parts of Alaska, central Greenland, Scandinavia and in the Russian Arctic (Novaya Zemlya archipelago) show the lowest average rate of SLA change. The mean rate of glacier

SLA change across grid cells within the *lowest* group is 0.7 m yr$^{-1}$ (SD=2.4 m yr$^{-1}$).

As for the Arctic–wide dataset, we also computed composite time series for grid cells within each group (**Fig. 6b and c**). For grid cells within the *highest* 25% of the distribution, a linear regression model (using the yearly mean values, i.e., **Fig. 6b**)



suggests a rise in SLA of 7.9±0.6 m yr$^{-1}$ (R$^2$=0.82, p<0.001), which corresponds to an average increase of 308 m (or 2.4 SD)
over the 1984–2022 period (given a 1 SD equivalence of ~129 m). While, for grid cells within the *lowest* 25% of the distribution
(**Fig. 6c**), a linear regression model suggests a much slower rise in SLA: 1.5±0.3 m yr$^{-1}$ (R$^2$=0.42, p<0.001). This corresponds
to an average increase of just 59 m (or 1.2 SD) over the 1984–2022 period (given a 1 SD equivalence of ~48 m). We note that
1 SD equivalences in meters were computed by taking the mean of the baseline SD for glaciers within each grid cell, and
across grid cells, for each group.


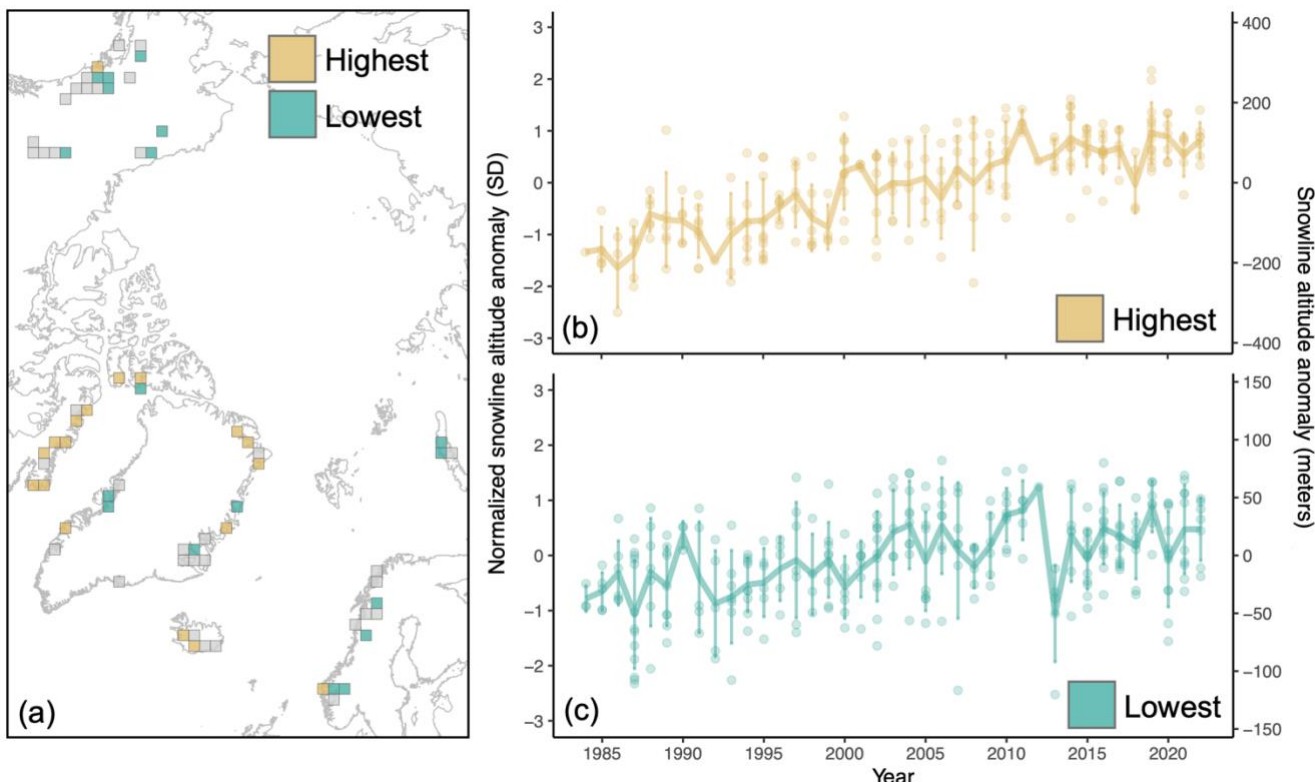

**Figure 6.** Quartile groups of grid cells that fall within the *highest* (yellow) and *lowest* (teal) 25% of the distribution of the rate
of snowline altitude change (see **Fig. S3**). (**a**) Spatial distribution of grids in the quartile groups. Map is in Lambert azimuthal
equal–area projection. Composite time series of snowline altitude from 1984–2022 for the mean of grid cells in (**b**) the *highest*
group, and (**c**) the *lowest* group. Uncertainty is represented as the spread, equivalent to 1 SD, of all grid cells within each group
(colored points) that contribute to the mean value for each year. Data are relative to the 1984–2022 mean. Note the different
right–side y–axes are based on a conversion of 1 SD=129 m in panel (b) and 1 SD=48 m in panel (c).

## 3.5 Changes in climate conditions and glacier–climate relationships



Fluctuations in mass balance of land–terminating glaciers are primarily controlled by two key climate variables, air temperature and precipitation (e.g., Dowdeswell et al., 1997; Oerlemans, 2005). Over the last four decades, Arctic summers (June, July, August) have warmed (M=0.04°C yr$^{-1}$; SD=0.02°C yr$^{-1}$), while trends in precipitation are more spatially variable (**Fig. 7**). At our studied glacier locations specifically, summers have warmed on average 1.2±0.04°C, over the 39 years of observation (**Fig. 7 and Table 2**). Given the average SLA rise of 152.1±2.5 m over the same period, this equates to a 127±5 m shift in SLA per

1°C of summer warming, disregarding the effects of precipitation. Comparatively, mean annual temperature has warmed 1.6±0.03°C, equating to a 95±2 m shift in SLA per 1°C of annual warming, again disregarding precipitation. However, since total annual snowfall has decreased and total annual rainfall has increased (–45 and +85 mm w.e., respectively; **Table 2**) on average at the glacier locations, this value should represent a maximum constraint on SLA sensitivity to temperature change, as the decrease in snowfall and increase in rainfall would act to enhance mass loss and SLA rise. Similarly to our study glacier

locations, total annual snowfall has decreased, and total annual rainfall has increased Arctic–wide. However, at our study locations and Arctic wide there is no significant trend in total precipitation change annually. This indicates that annually snowfall is decreasing not because it is overall drier, but because there are fewer freezing days as temperatures rise (i.e., the mode of precipitation is shifting to rain; **Table 2** and **Fig. 7**).

**Table 2.** Mean rate of annual and seasonal climate change at the study grid cells and Arctic–wide over the 1984–2022 period.

| | | Temperature (ºC yr$^{-1}$) | Snowfall (mm w.e. yr$^{-1}$) | Rainfall (mm w.e. yr$^{-1}$) | Total precipitation (mm w.e. yr$^{-1}$) | Freeze days (# of days yr$^{-1}$) | Positive degree day sum (ºC yr$^{-1}$) |
|---|---|---|---|---|---|---|---|
| **Grid cells mean** | *Annual* | **+0.04** | **–1.16** | **+2.18** | +1.02 | **–0.46** | **+3.11** |
| | *Summer* | **+0.03** | **–0.36** | **+1.12** | **+0.76** | **–0.21** | **+2.04** |
| | *Fall* | **+0.06** | –0.42 | **+0.69** | +0.27 | **–0.15** | **+0.65** |
| | *Winter* | **+0.04** | –0.38 | +0.09 | –0.28 | –0.01 | +0.01 |
| | *Spring* | **+0.03** | 0.00 | **+0.28** | +0.28 | **–0.08** | **+0.33** |
| **RGI mean** | *Annual* | **+0.06** | –0.34 | **+0.85** | +0.51 | **–0.44** | **+1.58** |
| | *Summer* | **+0.04** | **–0.37** | **+0.55** | +0.17 | **–0.35** | **+1.40** |
| | *Fall* | **+0.08** | +0.08 | **+0.25** | +0.33 | –0.07 | +0.13 |
| | *Winter* | **+0.09** | –0.08 | +0.01 | –0.07 | 0.00 | 0.00 |
| | *Spring* | **+0.05** | +0.03 | **+0.05** | +0.09 | **–0.02** | +0.05 |
| **Arctic mean** | *Annual* | **+0.06** | **–0.42** | **+0.76** | +0.35 | **–0.39** | **+5.25** |
| | *Summer* | **+0.04** | **–0.25** | **+0.30** | +0.05 | **–0.11** | **+3.37** |
| | *Fall* | **+0.08** | –0.20 | **+0.29** | +0.09 | **–0.16** | **+1.05** |
| | *Winter* | **+0.06** | –0.03 | +0.03 | +0.06 | –0.01 | +0.01 |
| | *Spring* | **+0.06** | 0.00 | **+0.14** | +0.14 | **–0.11** | **+0.81** |

Note: Data from ERA5−Land. Bold values indicate linear model is significant (*p<0.05; n=39*). RGI mean values include only glaciated areas between 60–90°N defined by the Randolph Glacier Inventory. Arctic mean values include all locations on land between 60–90°N.



**Figure 7.** Arctic–wide changes in climate over the 1984–2022 period. **(a)** Rate of summer temperature change (°C per year). **(b)** Rate of total annual snowfall change (mm w.e. per year). **(c)** Rate of total annual rainfall change (mm w.e. per year). **(d)** Rate of total annual precipitation change (mm w.e. per year). In all panels, the 70, 10,000 km² grid cell locations are denoted





as black boxes, and the 269 individual glacier locations are denoted as black points. Data is ERA5–Land reanalysis visualized on a 2,500 km$^2$ grid. Maps are in Lambert azimuthal equal–area projection.


Arctic–wide, we find moderate to strong positive correlations between glacier SLA and annual and seasonal temperature, with the strongest correlations between SLA and summer temperature, and SLA and annual temperature (0.74 and 0.73, respectively; **Fig. 8 and Table 3**). For precipitation, we find moderate negative correlations between glacier SLA and annual and summer snowfall (with the latter being the strongest; –0.63) and moderate positive correlations between glacier SLA and

annual and seasonal rainfall (with the former being the strongest; 0.67). Thus, although summer temperature generally controls glacier ablation, the strong positive relationship between annual temperature, annual PPD sum, and SLA highlights that warmer annual temperatures are contributing to both increased ablation (through enhanced melt and potential reduction of albedo due to increased rainfall) *and* reduced solid accumulation. The connection between increased annual temperature and reduced solid accumulation is also evident in the strong negative correlation between SLA and the number of freezing days annually (–0.77;

**Fig. 9**), which is indicative of the primary mode of precipitation that occurs (i.e., rain versus snow). We note that the relative weakness of correlations with precipitation as compared to temperature may be influenced by the coarse resolution of the ERA5–Land data and their inability to resolve local topographic effects on snowfall, rainfall, and total precipitation.

Consistent with the relationships based on non–detrended times series, the correlation among linearly detrended residuals is

statistically significant for SLA and summer temperature and for SLA and summer snowfall. In addition, the residuals of the detrended timeseries show a significant moderate negative correlation between SLA and total precipitation in summer, and a moderate positive correlation between SLA and spring rainfall. Further, among the linearly detrended residuals, there is a moderate negative correlation between SLA and annual and summer freeze days (with the latter being the strongest; –0.47), and a moderate positive correlation between SLA and annual and summer PPDs (with the former being the strongest; 0.37).

Overall, this indicates that both annual and summer season temperature, along with seasonal precipitation (and mode) are important climatic controls on both the overall rise in SLA and the variability in SLA between years. As such, positive SLA anomalies are generally associated with positive summer temperature anomalies, and with less snowy conditions in summer (**Fig. 10a**). A Welch Two Sample t–test suggests a significant difference in mean SLA anomaly between snowy (wet; –0.37 units of SD or –31 m) and non–snowy (dry; +0.28 units of SD or +24 m) summer years (t(37) = –3.89, p<0.001; assuming one

SD is equivalent to ~84 m; **Fig. 10b**).





**Figure 8.** Comparison between snowline altitude and climate variable composite time series (left) and detrended residuals (right) from 1984–2022. **(a, b)** Snowline altitude, **(c, d)** summer temperature, **(e, f)** summer snowfall, **(g, h)** annual rainfall (**i, j**) annual freeze days, and **(k, l)** annual positive degree days (PDDs). Time series are shown based on the strongest correlation with snowline for each climate variable in Table 3. Data are relative to the 1984–2022 mean. Absolute values are shown for freeze days and PDD sum.





**Table 3.** Zero–order correlations (*r*) between normalized composite time series of late–summer snowline altitude, and annual and seasonal climate variables.

| | Not detrended | | | | | | Detrended (residuals) | | | | | |
|---|---|---|---|---|---|---|---|---|---|---|---|---|
| | Temperature | Snowfall | Rainfall | Total precip. | Freeze days | PPD sum | Temperature | Snowfall | Rainfall | Total precip. | Freeze days | PDD sum |
| *Annual* | **0.73** | **−0.37** | **0.67** | 0.15 | **−0.77** | **0.75** | 0.22 | −0.26 | 0.12 | −0.15 | **−0.35** | **0.37** |
| *Summer* | **0.74** | **−0.63** | **0.57** | 0.18 | **−0.75** | **0.71** | **0.47** | **−0.53** | −0.01 | **−0.33** | **−0.47** | 0.33 |
| *Fall* | **0.64** | −0.21 | **0.44** | 0.06 | **−0.61** | **0.58** | 0.02 | −0.12 | 0.08 | −0.03 | 0.02 | −0.05 |
| *Winter* | **0.57** | −0.10 | **0.34** | −0.07 | −0.13 | 0.19 | 0.24 | −0.11 | −0.14 | −0.09 | −0.06 | 0.11 |
| *Spring* | **0.51** | 0.02 | **0.59** | 0.22 | **−0.51** | **0.47** | 0.20 | 0.03 | **0.33** | 0.10 | −0.19 | 0.04 |

Bold values are significant ($p<0.05$; $n=39$); PDD is the Positive Degree Day sum. We use absolute value composite time series for freeze days and PDD sum.

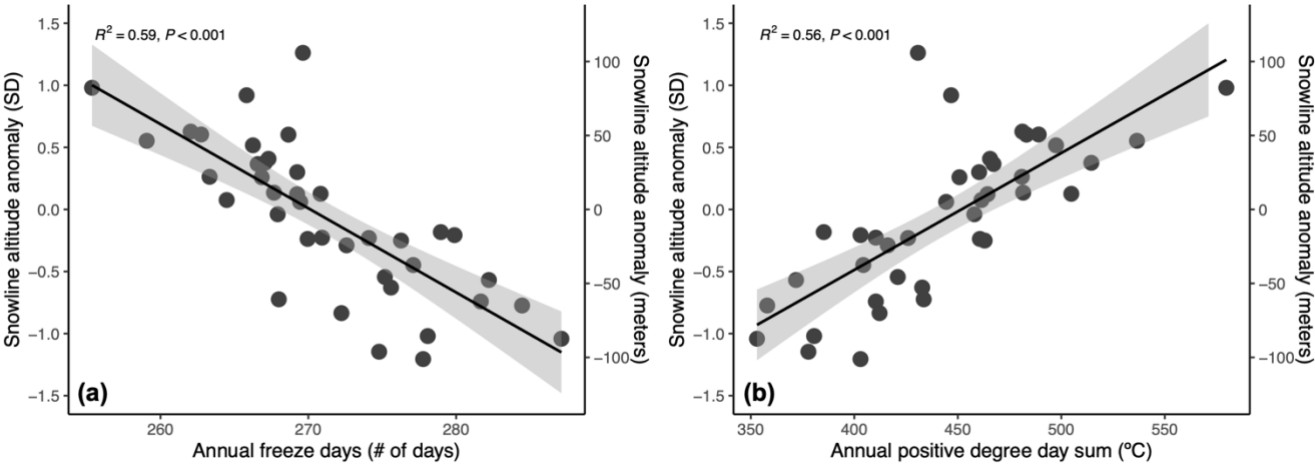

**Figure 9.** Relationship between snowline altitude and select temperature variables. **(a)** Normalized snowline altitude anomaly versus the number of days annually with a mean temperature ≤0 °C. **(b)** Normalized snowline altitude anomaly versus the annual positive degree day sum. Linear regression models along with 95% confidence intervals (gray bands) are overlaid in both panels.



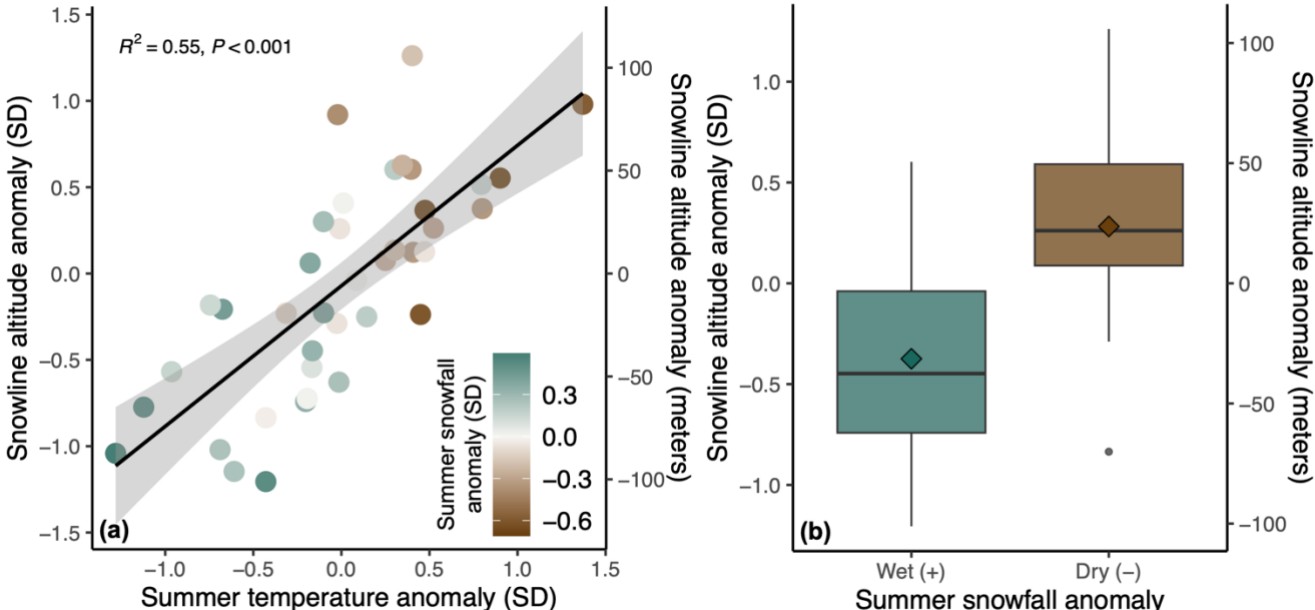

**Figure 10.** Relationship between snowline altitude and climate variables. **(a)** Normalized snowline altitude anomaly versus normalized summer temperature anomaly. Linear regression model along with 95% confidence interval (gray band) is overlayed. The color of the points reflects the normalized summer snowfall anomaly with drier (less snowy) years in brown and wetter (snowy) years in green. Data are relative to the 1984–2022 mean and each point represents one year in the 39–year period. **(b)** Box plots reflecting differences in the mean normalized snowline altitude anomaly (colored diamonds) for years with snowy (wet) and less snowy (dry) summers. In each box plot the center line represents the median, the edges of the box represent the first and third quartiles, and the whiskers extend to span a 1.5 interquartile range from the edges.

### 3.6 Glacier snowline altitude change and morpho–topographic variables

Considering each glacier individually, the rate of glacier SLA rise is significantly correlated with several glacier–specific morpho–topographic variables (**Table 4**). Specifically, we find a weak positive correlation between rate of SLA rise and glacier length and perimeter, indicating that SLAs are rising fastest on long and large glaciers. We also find weak to moderate negative correlations between the rate of SLA rise and glacier elevation and compactness, indicating that SLAs are rising slowest on compact, high elevation glaciers. For example, a characteristic glacier situated at a relatively low elevation (median elevation <1535 m; M=4.5 m yr$^{-1}$) is associated with a SLA rise that is about twice as fast as a characteristic glacier situated at a relatively high elevation (median elevation >1535 m; M=2.2 m yr$^{-1}$; Welch two–sample t–test, t(235)=5.46, P<0.001; **Fig. S4**).

**Table 4.** Zero–order correlations (*r*) between rate of glacier snowline altitude change and morpho–topographic variables.

| Morpho–topographic variable | |
|---|---|
| *Area* | *0.02* |
| *Length* | ***0.13*** |




| | |
|---|---|
| *Minimum elevation* | ***−0.37*** |
| *Median elevation* | ***−0.26*** |
| *Maximum elevation* | ***−0.16*** |
| *Slope* | *0.06* |
| *Aspect* | *0.04* |
| *Perimeter* | ***0.13*** |
| *Compactness* | ***−0.13*** |

Bold values are significant (*p<0.05; n=269*)

## 4 Discussion

### 4.1 Assumptions, limitations, and uncertainty

For some high–Arctic glaciers, superimposed ice formation represents an important control on mass balance (Woodward et al., 1997). Since superimposed ice can be difficult to distinguish from bare ice, especially in optical satellite imagery, and represents net accumulation occurring at elevations below the SLA, it can complicate mass balance and ELA determinations that rely on remotely observed methods (Bindschadler et al., 2001). However, it is generally established that in regions where measurements of glacier mass balance are limited, the SLA can still provide the best available estimate of ELA (e.g., Curley et al., 2021). In addition, Woodward et al. (1997) find that the extent and thickness of superimposed ice formation on High–Arctic glaciers are likely to be reduced with increased temperatures. Thus, we assume that the late summer SLA is roughly equivalent to the ELA, and do not take superimposed ice into account.

In addition, even though glacier SLAs were digitized as close as possible to the end of the ablation season, the maximum SLA was likely not captured for every year and glacier (e.g., about 68% of our observations of SLA were collected in the month of August; **Fig. S5**). However, there are a few available studies that give insight into ablation season TSL (i.e., SLA change within a summer season) and the timing of its maximum elevation. For example, Prantl et al. (2017) find that SLA on a high mountain Austrian glacier progressively rose over the 2014 summer season (from 2700 m up to 2960 m) but rose at a faster rate in early versus late summer, suggesting a steadying between August and mid–September. Using Landsat imagery, Mernild et al. (2013) determine the annual ELA for two Arctic glaciers between ~1999 and 2012 based on a second–order polynomial regression between the TSL elevation and the date of observation, where the ELA is defined as the highest value. They find consistency in the timing of the maximum transient SL elevation between years for Mittivakkat Gletscher in southeast Greenland, which occurred within a 4–day window in mid–August; while for Lemon Creek Glacier in southeast Alaska, the timing was more variable and occurred within a 27–day window in mid–August to mid–September (Mernild et al., 2013). Mernild et al. (2013) also find that the mean TSL elevation rate over the ablation season for these two Arctic glaciers was ~3.8 m per day and ~9.4 m per day for Lemon Creek and Mittivakkat, respectively.

Another limitation of our study is that it cannot account for deflating glacier surfaces, as we use a single DEM to extract the altitude of the glacier SLs over time. Since the ASTER GDEM V3 product was created from scenes acquired between the years 2000 and 2013, we suggest that SLAs before 2000–2013 should interpreted as minimum values, while SLAs after



2000–2013 should be interpreted as maximum values. And therefore, the overall rate of SLA rise is likely a maximum constraint. We also used the glacier surface elevation change dataset produced by Hugonnet et al. (2021), which covers the last two decades, to provide an estimate (maximum bound) on the cumulative elevation change for each glacier included in our analysis over our observational period (M=–27 m; **Fig. S6**). Thus, given this maximum bound on elevation change, and

that the majority of the surface deflation reported likely occurred below the SLA, we estimate that the effect of glacier surface deflation is unlikely to bias our estimates of SLA rise by more than ~1 m yr$^{-1}$. However, this represents a significant fraction of the overall rate of SLA change (~25% of ~3.9 m yr$^{-1}$).

Given that our overall approach does not focus on SLA change for individual glaciers, but rather on the average change for all study glaciers, we characterize uncertainty in our composite time series of SLA following Lorrey et al. (2022). Namely,

we quantify the variability or spread (i.e., 1 SD) for all grid cells that contributed to the mean normalized SLA value for a given year. The annual variability ranges from ±0.5 to ±1.1 SD, which is approximately equivalent to a range of ±42 to ±92 m. This approach differs from that of other studies that use interpolation methods to fill observational gaps and characterize uncertainty for each glacier and observation of SLA. For example, Rabatel et al. (2012, 2013) combine the following sources of independent error to characterize SLA uncertainty in the outer tropics and in the western Alps: the pixel size of the

imagery; the vertical accuracy of the DEM; the slope of the glacier in the vicinity of the SLA; and the SD of the average SL elevation along its delineation. This comprehensive approach resulted in a total SLA uncertainty ranging from ±11 to ±86 m in the outer tropics (Rabatel et al., 2012) and ±15 to ±170 m in the western Alps (Rabatel et al., 2013) depending on the year and glacier. These values are comparable to the spread in SLA across grid cells (i.e., ±42 to ±92 m) and suggests that our more generalized approach sufficiently represents uncertainty in our composite time series.

**4.2 Glacier snowline altitude change – comparison with other studies**

The Arctic–wide composite time series shows a clear trend, an overall rise in glacier SLA over the past 39 years, at a rate of 3.9±0.4 m yr$^{-1}$ (**Fig. 5b**). This value is broadly consistent with other reported rates of SLA and ELA change (inferred from field– and satellite–based observations) for glaciers in Arctic and non–Arctic regions of the world over recent decades (e.g., Ohmura and Boettcher, 2022; Rabatel et al., 2013; Lorrey et al., 2022). For example, while their field–based records of glacier

ELA cover a considerably longer time period, with a few beginning as early as 1945/46, Ohmura and Boettcher (2022) find a comparable shift in a global subset of glacier ELAs, ranging between +2 and +5 m yr$^{-1}$ (the average of the regional mean rate is +3.5 m yr$^{-1}$, and the average rate across all individual glaciers is +3.0 m yr$^{-1}$). From field observation in Arctic regions specifically, Ohmura and Boettcher (2022) also find similar spatial patterns in the rate of ELA rise: the fastest regional ELA rise occurred in Greenland (~7.6 m yr$^{-1}$) and in the Canadian Arctic (~7.5 m yr$^{-1}$), while slower ELA rise were observed in

Alaska (~6.3 m yr$^{-1}$), Norway (~4.3 m yr$^{-1}$), Sweden (~2.5 m yr$^{-1}$), and Iceland (~0.6 m yr$^{-1}$).





In addition, Rabatel et al. (2013) present time series of ELA derived from the end of summer SLA for 43 glaciers in the western Alps and report an average increase of ~170 m between 1984 and 2010 (or ~6.3 m yr$^{-1}$). In High Mountain Asia, Guo et al. (2021) delineate end of melt season SLAs based on albedo over 30 years (i.e., 1989–2019). The study finds an average SLA

increase of ~137 m (or ~4.6 m yr$^{-1}$) across the Altai and Karakoram mountains, and a higher average SLA increase of ~190–282 m (or ~6.3–9.4 m yr$^{-1}$) across the western Himalayas and Gongga Mountains (Guo et al., 2021). In the Southern Alps, Lorrey et al. (2022) present SLA time series for 41 glaciers between 1977 and 2020. The overall trend reflects an average rate of SLA increase of ~3.8 m yr$^{-1}$, however they suggest that much of the rise has occurred over the most recent three decades (Lorrey et al., 2022). In Arctic Canada, Curley et al. (2021) measure SLA on eight glaciers in eastern Ellesmere Island between

1974 and 2019 and report that the mean end of summer SL rose by ~360 m, or ~8 m yr$^{-1}$. However, this rate does not take interannual variability in SLA, nor the large temporal gaps in observation, into account (i.e., the rate is computed by taking the SLA of the last year of observation minus the SLA of the first year of observation and dividing by the number of years in the observational period; Curley et al., 2021).

## 4.3 Glacier sensitivity to climate variables

The mass balance of land terminating glaciers is well understood to be primarily controlled by summer season temperature and accumulated precipitation, however the characterization of collective glacier–climate relationships is complicated by their broad–ranging individual characteristics and local settings. This is evident in the variable response of individual glaciers to comparable climate forcings, emphasizing the unique sensitivities of these relatively small ice masses (e.g., McGrath et al., 2017; Carrivick et al., 2022, 2023). Their variable sensitivity also has implications for inferring the magnitude and regional–

scale spatial patterns of past climatic conditions from glacier ELA shifts. First, this implies that information from a large subset of glaciers is optimal, as glacier–specific factors could dominate the response of a single glacier or small sample (e.g., Rupper and Roe, 2008; Brooks et al., 2022; Carrivick et al., 2023). Second, ELA histories inferred from paleo–glacier reconstructions and used to constrain past climate rely on general assumptions of sensitivity to climate variables, the simplest of which prescribes the vertical shift in ELA to ablation season temperature using the atmospheric lapse rate (e.g., Denton et al., 2005;

Sikorski et al., 2009; Larocca et al., 2020a, b), while other glacier–climate relationships defined at the ELA relate a shift in its elevation to both temperature and precipitation (e.g., Ohmura et al., 1992; Joerin et al., 2008). Here, we assess our reported SLA changes in terms of sensitivity to changes in climate, and within this broader paleoclimate context.

Overall, we find that glacier SLA is most strongly correlated with the number of freezing days annually (–0.77), and strongly

correlated with the annual PPD sum (0.75; **Table 3**). In addition, when considering the multi–decadal global warming trend (non–detrended values), SLA is about as strongly correlated with annual temperature as it is with summer temperature. The strong relationships between SLA and annual climate variables probably reflect the effect of annual temperature on the amount of precipitation that falls as snow through the year. In other words, as annual temperatures rise, there are fewer days that reach freezing in which solid precipitation can fall and accumulate, and over decades, this results in a lengthening of the ablation

 

season. Whereas on inter–annual timescales (detrended values), the strongest relationships between SLA and climate variables

are seen in the summer season: namely snowfall, temperature, and the number of freezing days. This supports the notion that,

on inter–annual timescales, glacier SLA is controlled primarily by climate conditions during the ablation season. Interestingly,

among detrended residuals, we also find a moderate positive correlation between SLA and spring rainfall, indicating that

rainfall following winter, and prior to the ablation season, may act to raise end of summer SLAs, presumably through a

lowering of surface albedo.

Since the amount of solid accumulation has generally declined over the observational period at our study locations, we suggest

that our average estimated SLA sensitivity to summer temperature change, 127±5 m shift per 1 °C, represents a maximum

constraint. This estimate is roughly in the middle of the range of values for SLA sensitivity to temperature as reported in the

literature (~50–230 m per 1 °C) and is close to the standard environmental lapse rate (6.5 °C per 1000 m, or ~154 m per 1 °C)

used for some paleotemperature constraints. For example, in the Andes, Sagredo et al. (2014) examine the sensitivity of glaciers

to changes in temperature and precipitation and find ELA sensitivities ranging from ~140 to 230 m per 1 °C depending on the

climatic zone (however they note that due to model formulation, the relationship is governed by the local temperature lapse

rate). Other studies that focus on mid– to high–latitude glaciers find comparable ELA sensitivities. In the Tibetan Plateau,

Caidong and Sorteberg (2010) analyse the mass–balance sensitivity of Xibu Glacier and show that a temperature change of 1

°C shifts the ELA ~140 m. Oerlemans (1992) find that the climate sensitivity of glacier ELA in southern Norway range between

108–135 m per 1 °C. In the Alps, four studies find that the sensitivity of the ELA to air temperature lies within the range of

~115–130 m per 1 °C (Oerlemans and Hoogendoorn, 1989; Wallinga and Van De Wal, 1998; Gerbaux et al., 2005; Rabatel et

al., 2013), while a fifth finds a much lower ELA sensitivity to temperature, ranging from 50 to 85 m per 1 °C (Six and Vincent,

2014). Likewise, in Alaska and northwest Canada, McGrath et al. (2017) derive ELA sensitivities to climate for two glaciers

using historic data from the U.S. Geological Survey (USGS) Benchmark Glacier program and find ELA–temperature

sensitivities of 32–52 m per 1 °C. However, they note that these derived sensitivities are much lower than reported in other

published studies, and thus, that their projected ELA increases are likely conservative.

In general, we find that SLA is negatively correlated with annual and summer snowfall, and that snowfall has significantly

decreased annually and in the summer at the study locations over the 39 years of observation (**Table 2 and 3**). Additionally,

we find that SLA is positively correlated with annual and seasonal rainfall, and that rainfall has increased annual and seasonally

(except in winter) at the study locations over the 39 years of observation. Isolating the relative influence of changes in

precipitation from temperature on glacier ELA fluctuations remains a challenge, especially because annual temperature

impacts the number of freezing days, during which snow can fall. However, several studies have estimated the sensitivity of

the ELA to winter precipitation specifically, while others have assessed the percent increase in accumulated precipitation

needed to offset a 1 °C rise in temperature. Overall, most suggest that quite a large increase in accumulation season

precipitation is needed to offset a 1 °C warming. For instance, in the Andes, Sagredo et al. (2014) find that a ~33 to 290%



increase in annual precipitation is needed to compensate a 1 °C warming, and in France, Vincent (2002) finds that a 25–30%

increase in precipitation would compensate a 1 °C temperature rise. Other studies suggest that for most glaciers, a 30–50% increase in precipitation is required to cancel the effect of a 1 °C rise in temperature (Oerlemens, 2001; Braithwaite and Zhang, 2000; Caidong and Sorteberg, 2010). Quantitative estimates include Rabatel at al. (2013), who find an ELA sensitivity of ~48 m per 100 mm of winter precipitation and McGrath et al. (2017) who find an ELA sensitivity of +12 m per 10% decrease in solid precipitation. We note that the sensitivity of the ELA to increased rainfall is not as well understood.


It has also long been noted that the climatic setting and principal ablation process control the sensitivity of the glacier ELA to changes in temperature and precipitation (e.g., Braithwaite et al., 2002; Rupper and Roe, 2008; Anderson et al., 2010). In general, glaciers located in warm and wet maritime climate zones have a higher temperature sensitivity than those located in continental, dry and cold, settings (Braithwaite et al., 2002; Anderson et al., 2010). Further, Rupper and Roe (2008) show that

for glaciers in low precipitation settings, in which the main ablation process is sublimation, ELAs have a higher sensitivity to interannual variability in precipitation as compared to temperature. Conversely, for glaciers in high precipitation settings, in which the main ablation process is melt and surface runoff, ELAs have a higher sensitivity to interannual variability in temperature as compared to precipitation (Rupper and Roe, 2008). We split our study grid cells into "wet" and "dry" settings using the median value of the mean annual total precipitation over the entire observational period. In general, significant

correlations between detrended composite time series of SLA, temperature and snowfall, are stronger for the wet group than the dry group (**Table S2**).

**4.4 Glacier snowline altitude change and the role of morpho–topographic variables**

SLA change is significantly correlated with several morpho–topographic variables including glacier length, perimeter, elevation, and compactness (**Table 4** and **Fig. S7**). The rate of SLA change is negatively correlated with the latter two

variables, showing that SLA rise is slowest on compact glaciers at high elevations, presumably tucked in mountain cirques protected from insolation. The relationship between rate of SLA change, elevation and compactness, could be a result of more topographic shading in areas with steep mountain slopes. As glaciers retreat toward cirque headwalls, the overall amount of solar radiation received at the glacier surface is reduced (e.g., Olson and Rupper, 2019). Another contributing factor may be the negative feedback associated with orographic precipitation; as glaciers retreat to higher elevations, those higher elevations

have higher rates of precipitation, hypothetically countering retreat (e.g., Bolibar et al., 2022). High–elevation glaciated catchments may also benefit from the addition of wind–drifted and avalanched snow.

Our study focuses on SLA, which is an indicator of climate regionally, and is modified by local topographic controls, however observed changes in SLA should translate into glacier mass balance changes depending on the hypsometry of the glacier

surface. Several other studies have noted a relationship between glacier mass loss and elevation. For example, using the SLA method, Davaze et al. (2020) presented annual mass–balance time series for >200 glaciers in the European Alps between 2000–



2016 and find that steep, high elevation glaciers experienced less mass loss. The study also showed that 36% of the observed variance in glacier mass loss can be explained by three morpho–topographic characteristics: glacier slope, median and maximum altitude (Davaze et al., 2020). In addition, another study that focused on >1000 very small glaciers in the Swiss Alps 515 found highly variable mass balance sensitivities, with high–elevation, steep–sloped glaciers showing strongly reduced sensitivities to changes in temperature and precipitation, whereas low–elevation, gently–sloped glaciers were the most sensitive (Huss and Fisher, 2016). The enhanced sensitivity of glaciers with long tongues situated at lower and flatter terrain, is likely because a comparable rise in ELA would expose a greater proportion of their surface area to ablation.

## 4.5 Implications for the future

520

Future projections of glacier mass loss on global and regional scales remain uncertain, primarily because of glacier model uncertainty and emission pathway uncertainty, with the latter being the largest source of uncertainty by the end of the twenty–first century (Marzeion et al., 2020). Likewise, a recent novel study that presents a set of 21st century projections for every glacier on Earth, clearly shows that glacier mass loss is linearly related to temperature increase, implying that the magnitude 525 of temperature rise this century will have significant consequences for Earth's glaciers in terms of their mass loss and contribution to sea level rise (Rounce et al., 2023). This study projects that even if global mean temperature is limited to +1.5 °C above preindustrial levels, roughly half of the world's glaciers will be lost by the end of the century (Rounce et al., 2023).

Our study offers a simple, yet complementary approach to previous studies that have projected glacier loss (e.g., Rounce et al., 2023). Given our observations of SLA change we assess the future vulnerability of the studied glaciers by computing an 530 estimate of their end of century status:

$$SLA_{2100} = SLA_{ref} + (Grid_{rate} * 100), \tag{3}$$

where $SLA_{2100}$ is the estimated SLA position at the end of the twenty–first century (2100), $SLA_{ref}$ is the mean SLA during the reference period (1984–2022), and $Grid_{rate}$ is the mean rate of SLA change for the grid cell that each glacier falls within. Because the reference period for our SLA observations represents the multi–decadal average centred around the end of the 20th 535 century, we multiply the rate of SLA change (i.e., $Grid_{rate}$) by 100 years to estimate the SLA at the end of the 21st century. For each glacier and for all glaciers within each grid cell, we then compared the estimated SLA position at 2100 with the maximum ice elevation (**Fig. 11**). This gives an estimate of end of century glacier status—those with SLAs above the maximum ice elevation can be considered doomed to disappear as the glacier will no longer have an accumulation area. We find that out of the 269 studied glaciers, 141 (or ~52%) will have SLAs that exceed the maximum ice elevation by 2100 if the rate of change 540 remains consistent. The mean grid cell values show that SLA exceeds the mean maximum elevation for 38 (or ~54%) of the 70 grid cells, indicating grid cell locations with glaciers most at risk of extinction (**Fig. 11**).

We acknowledge that there are several assumptions and limitations of this approach. First, we assume that the mean rate of SLA change computed for each grid cell (i.e., **Fig. 5a**) remains constant, which is unlikely if temperature rise accelerates, and



solid accumulation continues to decline. In addition, we identify a negative feedback associated with changing morpho–
topographic setting that could slow SLA rise and mass loss as glaciers retreat to higher topographies. However, this effect
should be minor relative to the effect of warming and reduction in solid precipitation on glacier mass balance. Thus, we suggest
that these estimates can be considered conservative given continued amplified warming and projected reductions in Arctic
snowfall (in Arctic regions, rain is projected to become the dominant form of precipitation by the end of the century; e.g,
Bintanja et al., 2017; Rantanen et al., 2022).

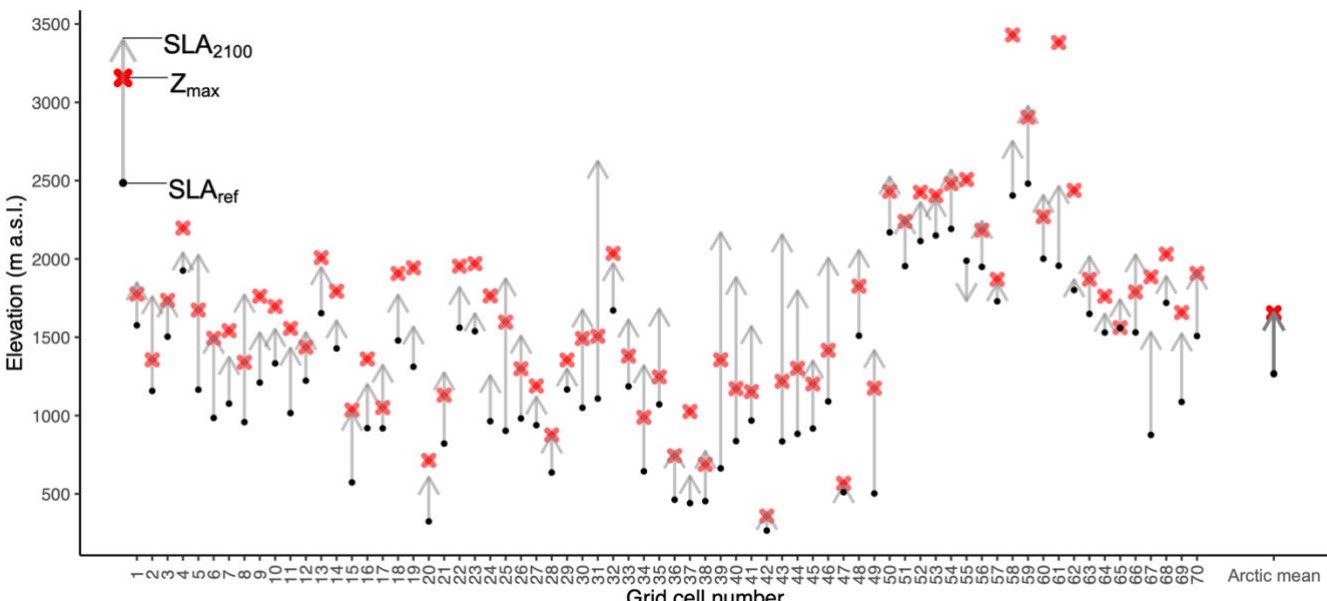


**Figure 11.** Assessment of glacier status in 2100 assuming that the mean rate of recent snowline altitude rise is maintained.
Arrow tops show the mean snowline altitude position at 2100 relative to the mean maximum ice elevation ($Z_{max}$; red cross)
grouped by grid cell. Black points represent the mean snowline elevation during the reference period (1984–2022). Arrow tops
that rise above the red crosses indicate a mean snowline rise above the mean maximum ice elevation such that the accumulation
area is lost. The Arctic mean values are shown on the right and assume the average rate of snowline altitude rise of 3.9 m yr$^{-1}$.

## 5 Summary

The regular monitoring of change in Earth's broadly dispersed and diverse glaciers remains an enormous challenge. Here,
using the suite of Landsat satellites, we measure change in late summer SLA for 269 glaciers in the Arctic over four decades.
We also compare the glacier SLA observations with long–term, field–based ELA measurements, ERA5–Land reanalysis
climate data, and with glacier specific morpho–topographic characteristics. We find that Arctic glacier SLAs have risen at an
average rate of 3.9±0.4 m yr$^{-1}$ between 1984 and 2022, and that a 1 ℃ rise in summer temperature corresponds to a ~127±5



m rise in SLA (not accounting for the effects of changing precipitation). We show that SLA is most strongly correlated with annual temperature variables, namely with the number of freezing days annually, and the annual PPD sum. When considering the multi–decadal warming trend, SLA is as strongly correlated with annual temperature as it is with summer temperature. This highlights the interconnected effect of annual temperature rise on both accumulation and ablation processes, with higher magnitude temperatures causing increased melt and fewer days in which solid precipitation can fall and accumulate. The observational evidence presented in this study also incorporates the effects of local physiography, and feedbacks related to the dynamics of shrinking ice (retreat to higher elevation), which are difficult to simulate. Specifically, we identify a significant morpho–topographic feedback on glacier retreat and show that SLAs on low elevation glaciers are rising about twice as fast as those on high elevation glaciers. Finally, assuming that the mean rate of SLA change remains constant, we find that roughly half of the glaciers studied here will be entirely below the local SL by the end of the century.

**Data Availability**

The data generated in this study (i.e., the raw glacier snowline altitude observations) are archived and publicly available at The Arctic Data Center: **DOI TBD**. Other supporting data are publicly available from the following sources: ERA5–Land reanalysis climate data (monthly averaged and hourly) can be found at the Climate Data Store: https://cds.climate.copernicus.eu/; glacier specific morpho–topographic data is available from the Randolph Glacier Inventory version 6.0 (RGI6.0) at https://nsidc.org/data/nsidc–0770/versions/6; and the ASTER GDEM is available at: https://search.earthdata.nasa.gov/search/?fi=ASTER.

**Author Contribution**

L.J.L., D.S.K, and J.M.L. designed the study. L.J.L and M.P. collected the snowline altitude measurements and L.J.L. conducted accompanying analyses. L.J.L and M.P.E. extracted, compiled, and performed analyses on all other data (climate and morpho–topographic variables). L.J.L. and D.S.K wrote the manuscript. L.J.L., D.S.K, J.M.L., M.P.E., N.P.M., M.P., and K.A.L. contributed to the discussion of the results, editing and revision of the manuscript.

**Competing Interests**

The authors declare that they have no conflict of interest.

**Acknowledgments**

This project was funded by the NOAA Climate and Global Change Postdoctoral Fellowship Program administered by UCAR's Cooperative Programs for the Advancement of Earth System Science (CPAESS) under award #NA18NWS4620043B to L.J.L, and by an undergraduate research grant from Northern Arizona University to M.P. We thank Yarrow Axford, Ben Phillips, Leah Marshall, Chris Hancock, Caitlin Walker, Hunter Allen, Maurycy Żarczyński, Allison Cluett, David Edge, Scarlett Hunt, and Franklin Telles who provided helpful comments and discussion.





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
