# Peer review of "Arctic glacier snowline altitudes rise 150 meters over the last four decades"

_EGUsphere, 2024_

## Referee Comment (RC1)

**Manuscript ID: TC-2024-522**

**https://doi.org/10.5194/egusphere-2024-522**

**Title:** Arctic glacier snowline altitudes rise 150 meters over the last four decades

**Authors:** Laura J. Larocca, James M. Lea, Michael P. Erb, Nicholas P. McKay, Megan Phillips, Kara A. Lamantia, Darrell S. Kaufman

**Reviewed by:** Dr. Antoine Rabatel (Univ. Grenoble Alpes)
**April 3, 2024**

**General comments**

The work by Laura J. Larocca et al. focuses on satellite–based observations of the glacier end-of-summer snowline altitude (SLA) as a proxy of the equilibrium-line altitude (ELA) on glaciers in the Arctic.

The snowline was mapped on a subset of 269 land–terminating glaciers above 60 °N latitude from Landsat satellite images between 1984 and 2022. The mean elevation of the snowline is extracted from the ASTER-GDEM.

The remotely–observed SLAs are compared with available in situ measurements of ELA, and the spatio-temporal changes are analyzed with regard to ERA5–Land reanalysis climate data. Overall, Arctic glacier SLAs have risen an average of ~150 m over the ~40-yr period. In parallel, a summer temperature shift of +1.2 °C at the glacier locations is quantified from the reanalysis data. This temperature increase goes hand-to-hand with an overall decrease in snowfall, an increase in rainfall and a decrease in the total number of freezing days.

Extrapolating the current rates of SLA rise, the authors finally show that half of the glaciers in the study regions could be below the ELA by 2100, and thus doomed to disappear.

All in all, I really enjoyed reading this article, whose theme and tools are particularly familiar to me. I think the work is very complete with figures (including the supplementary material) that are well done and used.

A large number of questions came to my mind during the reading (for example: the superimposed ice, the fact of considering only one DEM when the thickness of the glaciers has decreased over time, etc.), but the vast majority of them are dealt with in the discussion in an objective and clear manner, so I have few general comments on this work.

**1) Snowline *vs.* firn line. P4_L97-98**

Apparently, no distinction was made between the snow line and the firn line.

This is surprising at first sight, as for years with very negative mass balance it is often possible to distinguish between the two (including in optical data such as Landsat). Automatic algorithms have difficulty differentiating between the two, but for work based on manual digitization, and therefore on case-by-case visual expertise, experience shows that it is possible to differentiate between the two.

It would be interesting to see, for example for the glaciers on which the ELA is measured in the field, whether distinguishing between the snow line and the firn line (in years when both are visible) changes the quality of the SLA *vs.* ELA relationships.

**2) Selection of the glaciers. P4_L103-104**

I was a bit surprised to see that in your sample you have kept glaciers whose maximum altitude is not high enough to allow you to have a snow line every year.

I can admit that when glacier data are aggregated by grid, the weight of missing data is potentially reduced, but I have the impression that this inhomogeneity in the data series can create biases, particularly in the analysis of spatial or interannual variability (less so in the long-term trend).

Of the 50,000 glaciers in the Arctic region, it would probably be fairly easy to find the same number as those you have used, but with maximum altitudes high enough to allow the SLA to be measured every year, as was done in the work by Davaze et al. 2020 in the European Alps.

At the very least, I think it is necessary to discuss the impact of this temporal heterogeneity in the available data (Fig. S1) on the analysis of the results.

**3) Length of the snowline. P5_L109**

The length over which the altitude of the snow line is measured can vary greatly from one year to the next depending on the glacier's hypsometry. And not systematically calculating the snow line elevation over the same length can create inhomogeneity in the data series. Furthermore, for a given year, the position of the snow line may be dependent on factors other than climatic factors, for example on the edges of the glaciers, in connection with local effects due to avalanche deposits or shading linked to surrounding walls. For these reasons, in Rabatel et al. 2005 (and following) as well as in the regional study by Davaze et al. 2020, we chose to measure the altitude of the snow line on the central part of the glacier: more or less X metres on both side of the central flow line of the glacier (X being related to the size of the glaciers in the study area). Without asking to recalculate everything, I think it would be relevant to test the sensitivity of the results to the method (for example on the 30 glaciers where ELA data are available). And discuss this in the "Discussion" section.

**4) ELA vs. SLA. Section 3.1 P8-9.**

In section 3.1, you indicate that the SLA observed on satellite images may underestimate the ELA measured in the field because the images do not necessarily date from the end of the hydrological year. According to Figure S5, a significant number of images date from the period July to mid-August, and therefore most likely have a low SLA, but do you have the exact dates of the field measurements, so that you can quantify the time difference with the date of the image used to measure the SLA? This would provide more precise quantitative information on the uncertainty associated with this point.

**5) Uncertainty related to the glacier thinning. P19-20_L378-387**

You quantify precisely the average (maximum bound) error in the rate of rise of the SLA due to the (not considered) glacier thinning (~1 m/year, *i.e.* 25% of the average rate).

I feel that this should appear both in the Abstract at the beginning of the article and in the synthesis presented in Section 5, because even if this is the upper limit of the error linked to thinning, given the way it is calculated, over a long period of time as is the case here, this source of error cannot be neglected.

**Specific comments**
- In many places (starting by the second sentence of the abstract or L60), you use the term "parameter" when the term "variable" should be preferred. Please check carefully.
- In a few cases, you use "glaciated" when "glacierized" have to be used (refer to Cogley et al., 2011, Glossary of glacier mass balance and related terms). Please check carefully.
- P1_L15: "equilibrium-line altitude" (the hyphen is missing). Check everywhere in the text.
- P1_L28: I think using SLA in the sentence "… entirely below the SLA by 2100" is ambiguous. I understand what you mean, but I think you should rather use "ELA" instead of "SLA" or at least mentioning something like "late summer SLA".
- P3_Fig1: You could add the RGI regions' limits on this map (like on FigS6) but cutting the polygons at 60°.
- P9_L205: you can add that the glaciers of the subset are also "higher for median and maximum elevation".
- P13_L266: the reference by Dowdeswell et al. is missing in the ref list.
- P15_L295-297: You mention: "Thus, although summer temperature generally controls glacier ablation, … annual PDD sum … are contributing to both increase ablation…". Well, overall, this is correct, but I wonder to what extent the "summer temperature" and "annual PDD sum" variables are not highly correlated (or even self-correlated) and therefore whether there is not a high degree of redundancy between them.
- P15_L304-309: You should add some discussion of the positive and significant correlation (for raw values and residuals) between SLA and spring rainfalls.
  In my view, more liquid precipitation in the spring, to the detriment of snow precipitation, generates a thinner snowpack, which will melt more quickly in late spring/early summer, resulting in an earlier snow/ice (or snow/firn) transition and, due to the albedo effect, greater mass loss in late summer (*i.e.* a higher ELA).
  I think it is important to mention this point. Especially as this point about spring precipitation has already been highlighted in several studies (*e.g.*, Réveillet et al., 2018, Bolibar et al., 2020).
- P22_L448-450: In line with my previous comment, you should here expand a bit the discussion related to the impact of rainfall in spring.
- The paper by Jiskoot et al. in the reference list is not quoted in the text. To be removed.

References quoted in the review (not already mentioned in the ref. list of the paper)

Bolibar, J., Rabatel, A., Gouttevin, I., Galiez, C., Condom, T., & Sauquet, E. (2020). Deep learning applied to glacier evolution modelling. The Cryosphere, 14(2), 565-584.

Réveillet, M., Six, D., Vincent, C., Rabatel, A., Dumont, M., Lafaysse, M., … & Litt, M. (2018). Relative performance of empirical and physical models in assessing the seasonal and annual glacier surface mass balance of Saint-Sorlin Glacier (French Alps). The Cryosphere, 12(4), 1367-1386.

---

## Author Comment (AC1)

We thank the two reviewers for their constructive and thorough reviews that have helped us to strengthen our work. We address all reviewer comments in detail below.

Please find the reviewer comments in black and our responses in blue.

**Reviewer Comment 1 (RC1)**

**Manuscript ID: TC-2024-522**

**https://doi.org/10.5194/egusphere-2024-522**

**Title:** Arctic glacier snowline altitudes rise 150 meters over the last four decades

**Authors:** Laura J. Larocca, James M. Lea, Michael P. Erb, Nicholas P. McKay, Megan Phillips, Kara A. Lamantia, Darrell S. Kaufman

**Reviewed by:** Dr. Antoine Rabatel (Univ. Grenoble Alpes)
**April 3, 2024**

**General comments**
The work by Laura J. Larocca et al. focuses on satellite–based observations of the glacier end-of-summer snowline altitude (SLA) as a proxy of the equilibrium-line altitude (ELA) on glaciers in the Arctic.
The snowline was mapped on a subset of 269 land–terminating glaciers above 60 °N latitude from Landsat satellite images between 1984 and 2022. The mean elevation of the snowline is extracted from the ASTER-GDEM.
The remotely–observed SLAs are compared with available in situ measurements of ELA, and the spatio-temporal changes are analyzed with regard to ERA5–Land reanalysis climate data.
Overall, Arctic glacier SLAs have risen an average of ~150 m over the ~40-yr period. In parallel, a summer temperature shift of +1.2 °C at the glacier locations is quantified from the reanalysis data. This temperature increase goes hand-to-hand with an overall decrease in snowfall, an increase in rainfall and a decrease in the total number of freezing days. Extrapolating the current rates of SLA rise, the authors finally show that half of the glaciers in the study regions could be below the ELA by 2100, and thus doomed to disappear.

All in all, I really enjoyed reading this article, whose theme and tools are particularly familiar to me. I think the work is very complete with figures (including the supplementary material) that are well done and used.
A large number of questions came to my mind during the reading (for example: the superimposed ice, the fact of considering only one DEM when the thickness of the glaciers has decreased over time, etc.), but the vast majority of them are dealt with in the discussion in an objective and clear manner, so I have few general comments on this work.

We extend our thanks to Dr. Antoine Rabatel for the thoughtful and insightful review of our work. As an expert in the field who has published work using similar tools in the tropics, the feedback has been particularly valuable to us.

**1) Snowline vs. firn line. P4_L97-98**
Apparently, no distinction was made between the snow line and the firn line. This is surprising at first sight, as for years with very negative mass balance it is often possible to distinguish between the two (including in optical data such as Landsat). Automatic algorithms have difficulty differentiating between

the two, but for work based on manual digitization, and therefore on case-by-case visual expertise, experience shows that it is possible to differentiate between the two.
It would be interesting to see, for example for the glaciers on which the ELA is measured in the field, whether distinguishing between the snow line and the firn line (in years when both are visible) changes the quality of the SLA vs. ELA relationships.

In most instances the distinction between snow and firn was not possible or unclear in the relatively coarse 30-m resolution Landsat images used in our study (as similarly noted in your 2013 study of changes in glacier ELA in the western Alps using similar methodology). As a result, it is possible that for some of the glaciers studied, the firnline was delineated instead of the snowline. We decided to use the lower resolution Landsat images because we were interested in investigating long-term changes in snowline altitude and although there are higher resolution imagery available for recent years, we chose instead to keep the image resolution consistent throughout our ~40-year observational period. In addition, we think that any errors introduced in our estimates of snowline change over time (due to misidentification of the snowline) should be small as we have averaged over many glaciers each year. Further, in general changes in the firnline smooth out shorter-term variations and show trends over longer timescales, which is the focus of our study.

We acknowledge that this additional uncertainty was not well explained. We revised L97-98 for clarity and have added the following to Discussion section 4.1, Assumptions, limitations, and uncertainty following Rabatel et al., 2013:

*"Further, we acknowledge that in most instances, the distinction between snow and firn is not possible or unclear in the relatively coarse 30-m resolution Landsat images used in our study (e.g., Rabatel et al., 2013). As a result, it is possible that for some of the glaciers studied, the firnline (the boundary between firn and ice) was delineated instead of the SL (the boundary between snow and firn or snow and ice). Misidentifications of the SL as the firnline would act to underestimate the ELA and to smooth out short-term year-to-year variations but should have little effect on trends over long timescales, which is the overarching focus of our study (e.g., Yue et al., 2021)."*

**2) Selection of the glaciers. P4_L103-104**
I was a bit surprised to see that in your sample you have kept glaciers whose maximum altitude is not high enough to allow you to have a snow line every year.
I can admit that when glacier data are aggregated by grid, the weight of missing data is potentially reduced, but I have the impression that this inhomogeneity in the data series can create biases, particularly in the analysis of spatial or interannual variability (less so in the long-term trend).
Of the 50,000 glaciers in the Arctic region, it would probably be fairly easy to find the same number as those you have used, but with maximum altitudes high enough to allow the SLA to be measured every year, as was done in the work by Davaze et al. 2020 in the European Alps.
At the very least, I think it is necessary to discuss the impact of this temporal heterogeneity in the available data (Fig. S1) on the analysis of the results.

Thank you for this comment. We agree that ideally it would be best to have a snowline digitized every year for each glacier. While digitizing the glacier snowlines, we noted cases in which the entire glacier looked ice–covered. This only occurred for ~21 of the filtered set of 269 glaciers, mostly in limited instances in the more recent imagery. This equates to a very small fraction of the total number of snowlines digitized (~58 instances compared to the 3,489 digitized snowlines; less than 2%). We decided to keep these glaciers in the analyses because we planned to aggregate data by grid (i.e., When is SLA is above the glacier there isn't a straightforward way to estimate its elevation, however since we include glaciers across elevation ranges that higher SLA should be recorded on the higher elevation glaciers for that year). However, we acknowledge that the exclusion of these data could minimally dampen

interannual variability and our long-term SLA change estimates (making the longer-term projections more conservative).

Unfortunately, yearly digitization of the snowline was not possible for the set of glaciers we chose for this study. We acknowledge that in other work, yearly data were prioritized (e.g., Davaze et al. 2020); however, we also note that our study set out to characterize glacier snowline over a much longer period: 39 years (our study) versus 17 years (Davaze et al., 2020). We acknowledge that this is a limitation of our study as currently designed and we plan to keep this in mind for development of any future work on this topic. We also think that it is fair to argue, as you mention, that the effects of missing data should be reduced as we have normalized and aggregated the SLA data by grid and across all grid cells. Further, correlations (Table 3) between normalized composite time series of late–summer SLA, and annual and seasonal climate variables are relatively strong, suggesting that our yearly SLA composite values are reflecting annual/seasonal climate. We note that one exception to this may be the year 2012, where the number of SLA observations were particularly low (n=8) and the normalized value stands out as unusually high in the long-term trend (Fig. 5b). To address this concern, we have added discussion of the potential impact of this temporal heterogeneity in the available data (e.g., Fig. S1) on the interpretation of the results to Discussion section 4.1, Assumptions, limitations, and uncertainty:

*"Given that clear–sky imagery with limited terrain shadowing was sometimes not available at the end of the ablation season, we were unable to digitize the SL every year for each glacier. We note that this limitation, namely the year-to-year inhomogeneity in the number of SLA observations (Fig. S1) could create biases, particularly in the analysis of spatial and/or interannual variability (while less so in the long-term trend). Specifically, we think these data gaps could act to increase variability in SLA between years. However, given that correlations between yearly SLA and climate variables are relatively strong (i.e., Table 3), we suggest that our data aggregation methods (see Section 2.2) have helped to reduce these potential biases. One exception may be the year 2012, where our SLA observations were particularly low (Fig. S1)."*

**3) Length of the snowline. P5_L109**
The length over which the altitude of the snow line is measured can vary greatly from one year to the next depending on the glacier's hypsometry. And not systematically calculating the snow line elevation over the same length can create inhomogeneity in the data series. Furthermore, for a given year, the position of the snow line may be dependent on factors other than climatic factors, for example on the edges of the glaciers, in connection with local effects due to avalanche deposits or shading linked to surrounding walls. For these reasons, in Rabatel et al. 2005 (and following) as well as in the regional study by Davaze et al. 2020, we chose to measure the altitude of the snow line on the central part of the glacier: more or less X metres on both side of the central flow line of the glacier (X being related to the size of the glaciers in the study area). Without asking to recalculate everything, I think it would be relevant to test the sensitivity of the results to the method (for example on the 30 glaciers where ELA data are available). And discuss this in the "Discussion" section.

This is a great point, thank you. Although we aimed to digitize the glacier SL across the entire width of the glacier, when possible, we did not account for changing lengths of the line or any edge effects: we simply took the mean elevation across each snowline.

We take your suggestion and have measured the altitude of the snowline at the central point (more or less) on each of the 30 glaciers with long-term ELA observations (*n=278* snowlines). We compared the SL point measurements with our SL polyline measurements and with the field-based ELA observations. Overall, we find no significant difference between the two methods of calculating the snowline elevation (according to a Wilcoxon signed-rank test; V=28137, p-value=0.6327). We appreciate this exercise in

thinking about methodological robustness for future work. We have added discussion of this in Section 2.1 as follows:

*"We do not account for changing lengths of the digitized snowlines or any edge effects, however we compare our SL polyline observations with SL point observations taken along the central flow line (e.g., Rabatel et al., 2005; Davaze et al., 2020) for the 30 glaciers with long-term observations and find the methods to be comparable."*

Please also see figures below.

[Figure]

Comparison of manuscript Figure 4 (panels a and b) for case when glacier snowlines are digitized as a polyline (top) and for case when glacier snowlines are digitized as single point along the center of the glacier (bottom).

[Figure]

Comparison between snowline observed as line versus as point

Example (a)  Example (b)  Example (c)

$y = -8.27 + 1.01\,x,\ R^2 = 0.99,\ P < 0.001$

Region

● Alaska
■ Arctic Canada North
◇ Greenland
△ Iceland
▽ Scandinavia
✳ Svalbard

Top panels show three examples of glacier snowlines digitized as a polylines and as single points along the center of the glacier. Bottom panel compares the extracted snowline elevation using the two methodologies.

**4) ELA vs. SLA. Section 3.1 P8-9.**
In section 3.1, you indicate that the SLA observed on satellite images may underestimate the ELA measured in the field because the images do not necessarily date from the end of the hydrological year. According to Figure S5, a significant number of images date from the period July to mid-August, and therefore most likely have a low SLA, but do you have the exact dates of the field measurements, so that you can quantify the time difference with the date of the image used to measure the SLA? This would provide more precise quantitative information on the uncertainty associated with this point.

This is an excellent idea however the exact dates of the field measured ELAs were not included in the data compilation published by Ohmura and Boettcher (2022). They include only the hydrological year of the observation (see supporting information at: https:// www.mdpi.com/article/10.3390/w14182821/s1). We do our best to quantify this uncertainty by comparing the magnitude of transient SLA change over the ablation season given by Mernild et al. (2013) for two Arctic glaciers to an average estimated time lag to the end of the summer (L199-192).

**5) Uncertainty related to the glacier thinning. P19-20_L378-387**
You quantify precisely the average (maximum bound) error in the rate of rise of the SLA due to the (not considered) glacier thinning (~1 m/year, i.e. 25% of the average rate). I feel that this should appear both in

the Abstract at the beginning of the article and in the synthesis presented in Section 5, because even if this is the upper limit of the error linked to thinning, given the way it is calculated, over a long period of time as is the case here, this source of error cannot be neglected.

Yes, we agree this error (as well as the direction of the bias) should be more prominently presented. As suggested, we have added the following text to the Abstract:

*"However, we note that the effect of glacier surface thinning could bias our estimates of SLA rise by up to ~1 m yr$^{-1}$, a significant fraction (~25%) of the overall rate of change, and thus should be interpreted as a maximum constraint."*

In addition, in the Summary (Section 5) we have added to the following sentence: We find that Arctic glacier SLAs have risen at an average rate of 3.9±0.4 m yr$^{-1}$ between 1984 and 2022 …

*"We find that Arctic glacier SLAs have risen at an average rate of 3.9±0.4 m yr$^{-1}$ between 1984 and 2022 (not accounting for glacier thinning which could add an additional error of up to ~1 m yr$^{-1}$, making the reported rate of SLA rise a maximum constraint), and that a 1 °C rise in summer temperature corresponds to a ~127±5 m rise in SLA (not accounting for the effects of changing precipitation)."*

**Specific comments**
- In many places (starting by the second sentence of the abstract or L60), you use the term "parameter" when the term "variable" should be preferred. Please check carefully.

We have changed the term "parameter(s)" to the term "variable(s)" throughout.

- In a few cases, you use "glaciated" when "glacierized" have to be used (refer to Cogley et al., 2011, Glossary of glacier mass balance and related terms). Please check carefully.

Thank you for noting this. We have changed "glaciated" to "glacierized" throughout.

- P1_L15: "equilibrium-line altitude" (the hyphen is missing). Check everywhere in the text.

We have added the hyphen in L15 and have checked for inclusion of the hyphen throughout.

- P1_L28: I think using SLA in the sentence "… entirely below the SLA by 2100" is ambiguous. I understand what you mean, but I think you should rather use "ELA" instead of "SLA" or at least mentioning something like "late summer SLA".

We have specified that we mean the "late summer SLA"

- P3_Fig1: You could add the RGI regions' limits on this map (like on FigS6) but cutting the polygons at 60°.

We have added RGI region limits to Figure 1 and have revised it also according to RC2 suggestions. Please see RC2 comment on Figure 1.

- P9_L205: you can add that the glaciers of the subset are also "higher for median and maximum elevation".

We edited the sentence as follows:

*"In terms of morpho–topographic characteristics, in general our subset of glaciers are larger and longer, are less steep, and are higher in elevation (minimum, median and maximum elevation) as compared to the average across all land–terminating Arctic glaciers in the RGI (n=50,490; Fig. S2)."*

- P13_L266: the reference by Dowdeswell et al. is missing in the ref list.

Thank you. We have added Dowdeswell et al. to the reference list.

- P15_L295-297: You mention: "Thus, although summer temperature generally controls glacier ablation, … annual PDD sum … are contributing to both increase ablation…". Well, overall, this is correct, but I wonder to what extent the "summer temperature" and "annual PDD sum" variables are not highly correlated (or even self-correlated) and therefore whether there is not a high degree of redundancy between them.

It is correct that summer temperature and annual PDD sum are strongly correlated (0.94). However, we are aiming to highlight that in contrast to summer temperature (which can only impact the magnitude of ablation season melt), the PDD sum is distinct in that as temperature rises above 0°C throughout the year, it not only effects ablation rates (including the duration of the melt season), but also the number of days in which precipitation can fall as snow. To make our point clearer we add the following sentence:

*"This offers observational evidence for the importance of annual temperature (in addition to the conventional dominance of summer temperature) in controlling glacier health, and for the field of paleoscience, suggests that other paleoclimate variables may be reflected in records of past glacier states."*

- P15_L304-309: You should add some discussion of the positive and significant correlation (for raw values and residuals) between SLA and spring rainfalls. In my view, more liquid precipitation in the spring, to the detriment of snow precipitation, generates a thinner snowpack, which will melt more quickly in late spring/early summer, resulting in an earlier snow/ice (or snow/firn) transition and, due to the albedo effect, greater mass loss in late summer (i.e. a higher ELA). I think it is important to mention this point. Especially as this point about spring precipitation has already been highlighted in several studies (e.g., Réveillet et al., 2018, Bolibar et al., 2020).

Thank you, this is a great suggestion. We have added the following sentences to the noted paragraph:

*"As such, more spring rainfall, to the detriment of snow accumulation, leads to a thinner snowpack that melts more quickly in the late spring and early summer. This results in an earlier transition from snow to ice (or snow to firn), and due to the albedo effect, greater mass loss and a higher SLA in late summer."*

*"The important role of precipitation mode and seasonality (as well as liquid water content at the glacier surface) has been noted in other studies (e.g., Réveillet et al., 2018; Bolibar et al., 2020)."*

- P22_L448-450: In line with my previous comment, you should here expand a bit the discussion related to the impact of rainfall in spring.

We have added to the following sentence to expand upon the impact of spring rainfall:

*"This indicates that liquid precipitation following winter, and prior to the ablation season, may act to enhance summer mass loss and rise of end–of–summer SLAs, presumably through thinning of the winter snowpack, earlier melt, and a lowering of surface albedo."*

- The paper by Jiskoot et al. in the reference list is not quoted in the text. To be removed.

Thank you. We have removed Jiskoot et al. from the reference list.

References quoted in the review (not already mentioned in the ref. list of the paper)

Bolibar, J., Rabatel, A., Gouttevin, I., Galiez, C., Condom, T., & Sauquet, E. (2020). Deep learning applied to glacier evolution modelling. The Cryosphere, 14(2), 565-584.

Réveillet, M., Six, D., Vincent, C., Rabatel, A., Dumont, M., Lafaysse, M., ... & Litt, M. (2018). Relative performance of empirical and physical models in assessing the seasonal and annual glacier surface mass balance of Saint-Sorlin Glacier (French Alps). The Cryosphere, 12(4), 1367-1386.

---

## Author Comment (AC2)

We thank the two reviewers for their constructive and thorough reviews that have helped us to strengthen our work. We address all reviewer comments in detail below.

Please find the reviewer comments in black and our responses in blue.

**Reviewer Comment 2 (RC2)**

Review of Larocca et al. 'Arctic glacier snowline altitudes rise 150 meters over the last four decades', The Cryosphere Discussions.

This paper uses satellite observations of Arctic glacier end of summer snowlines as a proxy for equilibrium line altitudes. Snowlines were mapped for 269 land-terminating glaciers from Landsat optical images between 1984 and 2022. Snow line altitudes were extracted from ASTER GDEM elevation information and then compared with field-based ELA measurements and ERA-5 reanalysis climate information. The authors relate their changes to summer warming, decreases in snowfall, and increases in rainfall. I liked the paper and think it will make a valuable contribution to the literature. The paper is substantive and rigorous, and both the quality of writing and figures is excellent. I had just a few questions and suggestions as I went through the manuscript and will list them here for inclusion by the authors into a revised manuscript.

Thank you. We thank anonymous reviewer 2 for their valuable feedback and insightful comments and questions that have helped us to improve our manuscript. We address all questions and suggestions below.

- 37, worth differentiating here how many of these 42 are in the Arctic (not many).

Thank you, this is a good point that will help to highlight the dearth of long-term observational evidence specifically for Arctic glaciers. The WGMS report did not note the locations of the 42 glaciers (updated annually) with >30 years of continuous records, so instead we have rewritten the sentence to make the same point about the ~60 WGMS 'reference glaciers' (~26 of which are located in Arctic regions):

*"Although glaciological mass–balance observations from ~490 glaciers have been collected and are available at the World Glacier Monitoring Service (WGMS), only ~60 glaciers worldwide are considered 'reference glaciers', having continuous records spanning more than 30 years (less than half of which are in Arctic regions; WGMS, 2021).*

- 40, there's another data source that is relevant but not mentioned here - longer term (multi decadal) records of glacier area (and sometimes) volume change from archival aerial stereophotogrammetry. Find and cite some of these studies.

We have added the following sentence to refer to these other data sources:

*"Although the satellite era has opened expansive new opportunities for glacier monitoring from space, records that integrate historical data that precede the twenty–first century and that span multiple decades, are still lacking (e.g., Bauder et al., 2007; Papasodoro et al., 2015; Geyman et al., 2022)."*

- 54-55, this sentence would be better placed in the previous paragraph.

We agree and have moved this sentence to the first paragraph.

- 62, can you specify what is meant by 'glacier morpho-topographic variables' here, as it's a bit of an unwieldy term. I don't really know what it means but it seems important for it to be known at this crucial part of the manuscript. Probably worth a mention of the representivity of using land-terminating glaciers only too, given that the larger mass losses tend to be a tidewater terminating glaciers.

Yes, thank you for pointing this out. We define as follows:

... "glacier morpho–topographic variables (i.e., characteristics that describe physical state and geographic setting)."

We chose to focus on land-terminating glaciers because they more directly respond to atmospheric climate variability (as compared to glaciers in marine settings). To make that more clear we have added that explanation to the sentence that describes our criteria for glacier selection:

"All measured glaciers fit within the following criteria: the glacier (1) terminates on land (and thus responds directly to atmospheric climate variability); (2) is in the Arctic (which we define as land area above 60 °N latitude); and (3) is not surging (or has no record of surging behavior)."

- Figure 1, I find the very light green shading difficult to see, can this be made a darker, more distinct colour? Likewise, the difference between a small black dot and a small pink dot with a thick black edge is also not distinct. What about solid black vs solid blue?

Thank you for pointing this out. We have revised Figure 1 as follows: (1) We have made the green glacierized regions darker; (2) We have made the glacier symbols more distinct using solid black and solid blue as suggested. We denote the study glaciers using large blue circles and denote the glaciers with long-term ELA observations using large black triangles. We also add the RGI region extents and numbers as requested in RC1. Please see revised figure and caption below.

[Figure]

**Figure 1.** Map of the Arctic and glacier locations. Glacierized area distinct from the Greenland Ice Sheet is shown in green; the locations of the 269 land–terminating glaciers included in this study for which snowlines were digitized and analyzed are denoted by blue circles; and the locations of the 30 glaciers with long–term annual observations of equilibrium–line altitude (ELA) are denoted by black triangles. The latitude 60°N is denoted by the thick black line and first–order Randolph Glacier Inventory (RGI) region extents are defined by the shaded gray polygons with pink edges. The RGI regions are numbered as follows: 01 Alaska; 02 Western Canada and US; 03 Arctic Canada North; 04 Arctic Canada South; 05 Greenland Periphery; 06 Iceland; 08 Scandinavia; and 09 Russian Arctic. Glacierized area and first–order regions are from the RGI Version 6 (Pfeffer et al., 2014; RGI Consortium, 2017).

- Section 2.1, presumably an elevational range is important. Can you summarise the range of elevations over which these 269 glaciers lie (maybe ELA elevation). I would think you need a range of low, medium and higher elevation sites. If you do not have this, please justify why not.

Yes, elevation range is important, and we look closely at glacier morpho-topographic variables (including minimum, median, and maximum elevation; Fig. S2 summarizes glacier elevation ranges). Although, in general, our selected glaciers are higher in elevation than the Arctic-wide average, we do have a range of glaciers spanning lower, mid, and higher elevations. We find that SLAs on lower elevation glaciers are rising roughly twice as fast as those on higher elevation glaciers (Section 3.6).

- 97, sorry, I'm a bit unclear about this - you used a digitisation tool, yet digitised all SLs manually? Specify clearly what the tool was for, and what the tool did not do.

No worries. Yes, the tool allowed us to digitize glacier snowlines directly in Google Earth Engine (i.e., provides on-the-fly access without the need for downloading of large images files). We attempt to make this clearer in the tool's description:

*"To digitize the position of the glacier SL, we used the Google Earth Engine Digitisation Tool (GEEDiT), which allows for rapid on–the–fly access to, and visualization of the full Landsat Tier 1 imagery collection, as well as rapid mapping of georeferenced vectors that can be exported with image metadata (Lea et al., 2018)."*

- 98, as ever with these remote sensing-based estimates of snow line / ELA, how do you account for the presence and influence of the superimposed ice zone? [note, I see that this is addresses in section 4.1]. And then further, why do you not differentiate between the snow line and the firn line which you may be able to distinguish in optical imagery in some years.

Unfortunately, we cannot account for the presence of the superimposed ice zone, however our comparison with field-based ELA observations suggests that this does not present a major issue for our remotely-sensed SLA observations (i.e., we would expect the remotely–observed SLAs to be systematically higher than the field–measured ELAs if superimposed ice zones were consistently missed in the optical imagery).

Regarding the distinction between the snowline and firnline, please see our response to RC1 (general comment 1).

- 103-104, why do you include glaciers whose maximum altitude is higher than the snowline? With your sample size, I'm sure it would be straightforward to exclude all those glaciers with such low maximum elevations and include only those that allow the snowline to be measured every year. Inclusion of these glaciers is likely to bias your results in some way, so as an absolute minimum you need to account for this and discuss it.

Thank you for this question. Please see our response to RC1 (general comment 2).

- 109, will mean snowline elevations be affected by factors such as changing glacier hypsometry (as the snowline changes) and avalanching onto the glacier surface? These may be relevant at local scales. If you can quantify, do, otherwise mention as a potential source of uncertainty.

Thank you for this comment. Local scale processes (microclimates, shading effects, avalanching, wind, and redistribution of snow, etc.) can have strong effects on glacier snowline (L 171-175). We try to account for these local factors on SLA by including comparison to several variables that describe glacier morphology (Section 2.3).

Glacier hypsometry should only have indirect effects on snowline (through the variables mentioned above). For example, a steeply sloped glacier with most of its area concentrated at high elevations might be more affected by avalanching. However, the distribution of the glacier surface area with altitude does affect how changes in SLA translate into changes in glacier mass balance.

Below we show that the relationship between the rate of SLA rise and glacier hypsometry is not significant. We quantified the hypsometric character of each glacier, using a hypsometric index (HI) as defined in Jiskoot et al. (2009):

$HI = (H_{max} − H_{med}) / (H_{med} − H_{min})$, and if $0 < HI < 1$, then $HI = −1/HI$,

where $H_{max}$ and $H_{min}$ are the maximum and minimum glacier elevation, and $H_{med}$ is the median elevation. We also then grouped the glaciers into five HI index categories defined by Jiskoot et al., (2009), where: *H < −1.5* is very top heavy; *−1.5 < H < −1.2* is top heavy; *−1.2 < H < 1.2* is equidimensional*; 1.2 < H < 1.5* is bottom heavy; and *H > 1.5* is very bottom heavy (Jiskoot et al., 2009).

We found that the relation between glacier SLA change and the HI index on an individual level is not significant. An ANOVA revealed statistically significant differences between the five HI categories in terms of mean rate of SLA change ($F_{(4, 263)}=3.46$, $P=0.009$). However, post-hoc comparisons using Tukey's HSD test showed that none of the differences between categories were significant ($P \geq 0.05$).

Reference: Jiskoot, H., Curran, C.J., Tessler, D.L. and Shenton, L.R.: Changes in Clemenceau Icefield and Chaba Group glaciers, Canada, related to hypsometry, tributary detachment, length–slope and area–aspect relations. *Annals of Glaciology*, *50*(53), pp.133-143, 2009.

- Table 1, spectral range of bands (if different)?

We used optical imagery only (RGB).

- Figure 3, why no glaciers in Svalbard which surely is the most well studied of these regions? But Svalbard is in the Figure 4 plot?

We did not place the same constraints as far as temporal coverage on the glaciers with long-term ELA measurements. In other words, none of the glaciers in Svalbard met our requirements for temporal coverage for inclusion in the subset of 269 glaciers in which we carried out the larger analyses with climate, etc. Our requirements are 5 or more total SL observations over the observational period; at least 1 observation in each third of the observational period; and a maximum gap of 15 years between SL observations).

- 128, Cite the RGI paper as well as the dataset (Pfeffer et al., 2014, doi: 10.3189/2014JoG13J176.

Thank you. We have added this citation to the reference list and cite it throughout the paper along with the RGI dataset reference.

- 183, *n*=?

*n=278.* We have added the number of SLA-ELA observations as follows:

*"We find a robust relationship between the remotely–observed SLAs and the field–measured ELAs for the 30 glaciers (n=278) with long–term, quality observations (**Fig. 4a**; $R^2=0.92$, $p<0.001$)."*

- 307, more could be said about the correlations between SLA and spring rainfall. I would expect spring rain to result in less snow due to melting from latent heat release and then enhanced rates of compaction and will likely lead to an earlier snow melt and then higher ELA.

Thank you, this is a great suggestion. We have added more discussion on spring rainfall. Please see our response to RC1 (comments on P15_L304-309 and P22_L448-450).

- 378, It would worth attempting to quantify the error in the rate of SLA rise due to glacier thinning and include it here and in the main summary of results. Also, 'deflating' here and elsewhere is an odd word choice. Suggest 'thinning', or 'lowering'.

We now include the maximum error (~1 m per year) due to glacier thinning in the Abstract and Summary. We also change the term "deflating" to "thinning" or "lowering" throughout.

- Dowdeswell ref in main text not in the ref list, and Jiskoot ref in list not in the main text

Thank you. We added Dowdeswell and removed Jiskoot from the main manuscript reference list.

**Citation**: https://doi.org/10.5194/egusphere-2024-522-RC2